# Exploration of sensory and spinal neurons expressing gastrin-releasing peptide in itch and pain related behaviors

Devin M. Barry [1,2], Xue-Ting Liu [1,2,3], Benlong Liu[1,2], Xian-Yu Liu[1,2], Fang Gao [1,2], Xiansi Zeng [1,2,6], Juan Liu [1,2], Qianyi Yang[1,2], Steven Wilhelm [1,2], Jun Yin [1,2], Ailin Tao [1,3] & Zhou-Feng Chen[1,2,4,5 ✉]

Gastrin-releasing peptide (GRP) functions as a neurotransmitter for non-histaminergic itch, but its site of action (sensory neurons vs spinal cord) remains controversial. To determine the role of GRP in sensory neurons, we generated a floxed *Grp* mouse line. We found that conditional knockout of *Grp* in sensory neurons results in attenuated non-histaminergic itch, without impairing histamine-induced itch. Using a *Grp*-Cre knock-in mouse line, we show that the upper epidermis of the skin is exclusively innervated by GRP fibers, whose activation via optogeneics and chemogenetics in the skin evokes itch- but not pain-related scratching or wiping behaviors. In contrast, intersectional genetic ablation of spinal *Grp* neurons does not affect itch nor pain transmission, demonstrating that spinal *Grp* neurons are dispensable for itch transmission. These data indicate that GRP is a neuropeptide in sensory neurons for non-histaminergic itch, and GRP sensory neurons are dedicated to itch transmission.

[1] Center for the Study of Itch and Sensory Disorders, Washington University School of Medicine, St. Louis, MO 63110, USA. [2] Department of Anesthesiology, Washington University School of Medicine, St. Louis, MO 63110, USA. [3] The Second Affiliated Hospital, The State Key Laboratory of Respiratory Disease, Guangdong Provincial Key Laboratory of Allergy & Clinical Immunology, Sino-French Hoffmann Institute, Center for Immunology, Inflammation, Immune-mediated disease, Guangzhou Medical University, 510260 Guangzhou, Guangdong, P.R. China. [4] Department of Psychiatry, Washington University School of Medicine, St. Louis, MO 63110, USA. [5] Department of Developmental Biology, Washington University School of Medicine, St. Louis, MO 63110, USA. [6]Present address: College of Life Sciences, Xinyang Normal University, 237 Nanhu Road, 464000 Xinyang, P. R. China. ✉email: chenz@wustl.edu

Modality-specific information is transmitted and processed through the superficial dorsal horn of the spinal cord. Primary afferents of sensory neurons release fast neurotransmitters (e.g. glutamate) and neuropeptides to activate postsynaptic receptors in the spinal cord dorsal horn to transmit itch and pain information[1–4]. Previous evidence suggested that gastrin-releasing peptide (GRP) is a principal neuropeptide for transmitting non-histaminergic itch from primary afferents to GRP-receptor (GRPR) neurons in the dorsal horn[1,5–9]. The potential evolutionary role for GRP was supported by the finding that GRP in the suprachiasmatic nucleus of hypothalamus is crucial for contagious itch behavior[10] and by its conserved expression in the dorsal root ganglion (DRG) neurons between rodents and primates[8,11–13], and more recently in dogs[14]. In contrast, neuromedin B (NMB), the second mammalian bombesin-related neuropeptide[15], relays histamine-evoked itch from DRGs to the dorsal horn interneurons expressing NMB receptor (NMBR)[6,16]. However, there are considerable disagreements whether $Grp$ is expressed in DRGs[7,17–21], due to its relatively low copy number of mRNA. Although the presence of $Grp$ mRNA in DRGs has been well described, the direct evidence for the role of GRP is still lacking. Addressing this question necessitates the generation of floxed $Grp$ mice for DRG-specific knockout (KO). On the other hand, attracted by abundant $Grp$ mRNA expression in the spinal cord[19,22], spinal $Grp$ neurons have been postulated to act either as a dedicated neural station that sends itch information to GRPR neurons via release of GRP[17,23,24], or a "leaky gate" for itch and pain[25]. These studies, however, employed GENSAT's BAC transgenic $Grp$-Cre/eGFP mouse lines which capture only approximately 25% of endogenous $Grp$ expression in the spinal cord[18,24,26], indicating a lack of utility of BAC $Grp$ mouse lines for functional interrogation of spinal $Grp$ neurons in itch and pain. A critical appraisal of the function of $Grp$ neurons in the spinal cord using intersectional genetic ablation combined with the gold standard itch behavioral assay entails a generation of a high-fidelity $Grp$-Cre mouse line using knock-in approach.

In this study, we generated and validated a long-sought knock-in allele in which Cre recombinase (Cre) is driven under the control of the endogenous $Grp$ promoter ($Grp^{Cre-KI}$) and a floxed $Grp$ allele for conditional deletion of $Grp$ in DRGs. Using these mouse toolkits, we investigated the role of $Grp$ in DRGs, GRP primary afferents and spinal $Grp$ neurons in itch and pain.

## Results

**$Grp^{Cre-KI}$ expression in sensory neurons and skin fibers.** To investigate the role of $Grp$ sensory neurons in itch, a targeting construct was generated with an eGFP-Cre recombinase cassette and knocked-in at exon 1 start codon of $Grp$ ($Grp^{Cre-KI}$) in mice (Fig. 1a, b). eGFP-positive neurons in DRG were detected following immunohistochemistry (IHC) with a GFP antibody, but epifluorescent eGFP signals were not observed (Fig. 1c, d). Moreover, the GFP antibody did not detect any signals in $Grp^{WT}$ tissues (Supplementary Fig. 1a). To validate specific expression of $Grp$ in DRG neurons, we employed RNAScope™ in situ hybridization (ISH) methods[27] and found that $Grp$ was detected in a subset of wild-type (WT) DRG neurons (~7%) but not in $Grp$ KO tissues (Supplementary Fig. 1b, c). This result is consistent with $Grp$ mRNA (~7%) detected by conventional digoxygenin (DIG) (Supplementary Fig. 1d) as well as previous studies[7,19]. To verify the activity of Cre recombinase in adult DRGs, Cre-dependent eYFP virus (AAV5-EF1α-DIO-eYFP) was injected into the DRGs of adult $Grp^{Cre-KI}$ mice. Three weeks after virus injections, eYFP was expressed in DRG neurons, which also expressed GRP (Fig. 1e–g). These results indicate that Cre recapitulates the

expression of GRP in DRG. Previously we found that the signal of GRP immunostaining was significantly reduced in the SCN after mice watched scratching video, suggesting that GRP was released to activate GRPR neurons to mediate contagious itch behavior[10]. We speculated that if GRP functions as an itch peptide in DRGs, it should also be released into the spinal cord in response to intradermal injection (i.d.) of a pruritogen, resulting in reduced immunostaining of GRP fibers. To test this, we performed GRP antibody staining in the cervical spinal sections following i.d. injections of chloroquine (CQ), a non-histaminergic pruritogen that is mediated by GRP-GRPR signaling[5]. The GRP immunostaining signals on the ipsilateral side of CQ injection were significantly reduced compared to the contralateral side ($p = 0.017$) (Fig. 1h). In contrast, substance P (SP) IHC signaling were similar between two sides of the cervical spinal cord following CQ injections ($p > 0.99$) (Fig. 1i). These results suggest that GRP in the primary afferents was released upon itch stimulation.

To characterize innervation of $Grp$ sensory neurons in the skin, $Grp^{Cre-KI}$ mice were crossed with the tdTomato flox-stop reporter line (Ai14)[28] to generate mice with tdTomato expression in $Grp^+$ neurons ($Grp^{tdTom}$)(Fig. 2a). To validate tdTomato expression, we performed IHC of eGFP with tdTomato epifluorescent signals in $Grp^{tdTom}$ DRGs. tdTomato was expressed in nearly all DRG neurons that expressed eGFP in $Grp^{tdTom}$ mice (98/99 tdTom/eGFP and 98/102 eGFP/tdTom neurons) (Fig. 2b, c). IHC of GRP in DRG of $Grp^{tdTom}$ mice confirmed that tdTomato epifluorescent signals are colocolalized with GRP (Fig. 2d–f). We also validated the $Grp^{Cre-KI}$ expression in the spinal cord by $Grp$ ISH in $Grp^{tdTom}$ spinal sections and found that 89% of tdTomato$^+$ neurons expressed $Grp$ mRNA, and, importantly, 93% of $Grp$ mRNA$^+$ neurons expressed tdTomato (Supplementary Fig. 2c–e). Of note, GRP antibody labeled neuronal terminals in the superficial dorsal horn but not cell bodies (Fig. 2g–i). Importantly, GRP IHC staining in the spinal cord exhibits a characteristic feature that resembles those neuropeptides originated in DRGs such as CGRP: the most intense staining is present in the most superficial layer, with gradual reduction onto lamina II. Such a pattern does not fit with the notion that GRP is released from endogenous spinal $Grp$ neurons, which, if they function only through GRPR neurons directly, should exhibit laminae I–II restricted homogenous IHC pattern. Intriguingly, while $Grp^{tdTom}$ cells were present in DRGs and innervate the skin broadly (see below), we did not detect central terminals marked by tdTomato in the spinal cord (Fig. 2g). The lack of tdTomato-labled central terminals has also been shown in other Cre lines, such as $Tac1^{Cre}$-tdTomato mouse line[29]. The reason for this discrepancy is currently unknown. Nevertheless, tdTomato expression was not detected in the DRG and skin of $Grp^{WT}$;Ai14 mice (Supplementary Fig. 2a, b). The results indicate that tdTomato expression faithfully recapitulates $Grp$ expression and can be used as a surrogate for $Grp$ fibers.

To characterize the innervation organization of GRP fibers in the skin, we examined a wide array of tissues and found that they received innervation from $Grp^{tdTom}$ fibers with a focus on the epidermis and cutaneous structures (Supplementary Table 1). $Grp^{tdTom}$ was observed in nerve fibers of hairy nape skin, with apparent co-expression in some fibers expressing calcitonin gene-related peptide (CGRP), a peptidergic marker of nociceptive neurons (Fig. 2j, arrows)[30]. Some $Grp^{tdTom}$ fibers that were CGRP-negative wrapped around the upper epidermal regions of hair follicles as apparent circular or penetrating follicle neck endings (Fig. 2j, asterisk). $Grp^{tdTom}$ fibers showed apparent intertwining with CGRP fibers or with some fibers showing co-expression. Next, we examined the thicker glabrous skin of the paw to better visualize the termination zones of $Grp^{tdTom}$ fibers. Many $Grp^{tdTom}$ fibers projected from the dermal/epidermal

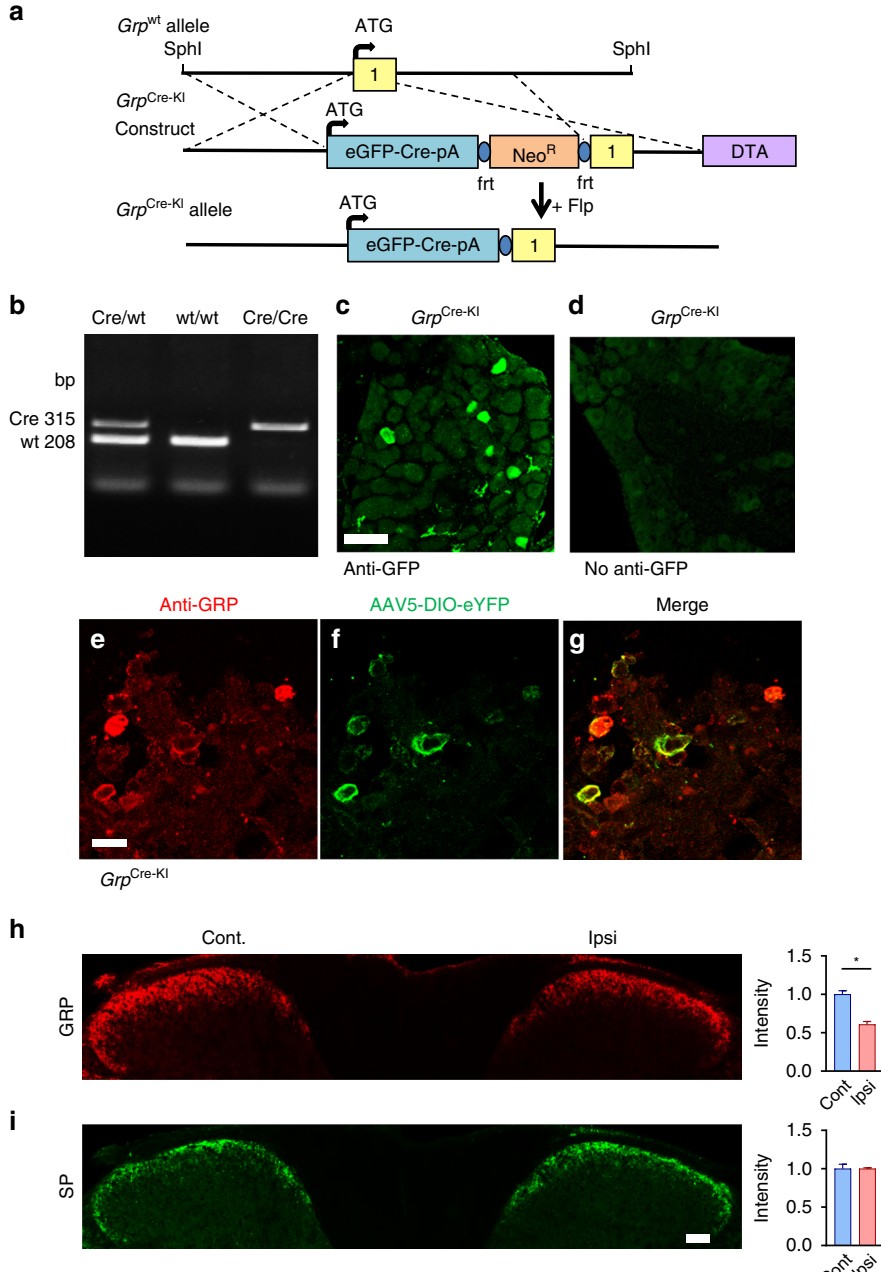

**Fig. 1 Validated expression of *Grp* and *Cre* in *Grp*[Cre-KI] sensory neurons. a** Schematic of targeting strategy to knock in eGFP-Cre cassette in *Grp* allele to generate *Grp*[Cre-KI] mice. **b** Gel electrophoresis of genotyping PCR from *Grp* Cre/wt, wt/wt, and Cre/Cre samples. **c** eGFP IHC from DRG section of an adult *Grp*[Cre-KI] mouse. **d** Image of eGFP epifluorescent signal (no GFP antibody) in *Grp*[Cre-KI] DRG. **e–g** GRP IHC (**e**) and eYFP fluorescent image (**f**) from DRG section of an adult *Grp*[Cre-KI] mouse that received AAV5- EF1α-DIO-eYFP injection into the DRG for 3 weeks. Scale bar, 20 μm. **h** and **i** IHC of GRP (**h**) and SP (**i**) in cervical spinal sections from WT mice following i.d. injections of CQ (200 μg). Scale bar, 100 μm. Data are presented as mean ± s.e.m., *n* = 3 mice and 9 sections. Source data are provided as a Source Data file.

boundary, mostly straight up through the stratum basilis and stratum spinosum, but then meandered and wrapped around keratinocytes in apparent 'S' or 'Z' pattern free endings (arrows) or as bush/cluster endings (arrowhead) within the stratum granulosum and terminated ~5 μm from the stratum corneum (Fig. 2k, red). In contrast, most of the CGRP fibers along with a few *Grp*[tdTom] fibers (asterisk) were straight with less complexity and terminated mostly in the stratum spinosum (Fig. 2k, green). *Grp*[tdTom] was not co-expressed with myelinated fibers using an antibody against neurofilament heavy (NF-H)(Supplementary Fig. 2f–h)[31,32], and no *Grp*[tdTom] fibers were observed in any

sensory structures that are typically innervated by myelinated fibers including Merkel cell-complexes, Meissner corpuscles, vibrissa follicle-sinus complexes, or sebaceous glands (Supplementary Table 1). *Grp*[tdTom] fibers were also absent from the heart, esophagus, intestine, kidney, liver, lung, and testis (Supplementary Table 1). *Grp*[tdTom] fibers were rarely observed in the epithelial layers of the tongue, cornea, and bladder (Supplementary Fig. 2i–k). *Grp*[tdTom] fibers were also barely present in skeletal muscle within coursing nerve bundles with CGRP fibers but did not innervate muscle fibers or the neuromuscular junction (Supplementary Fig. 2l, m).

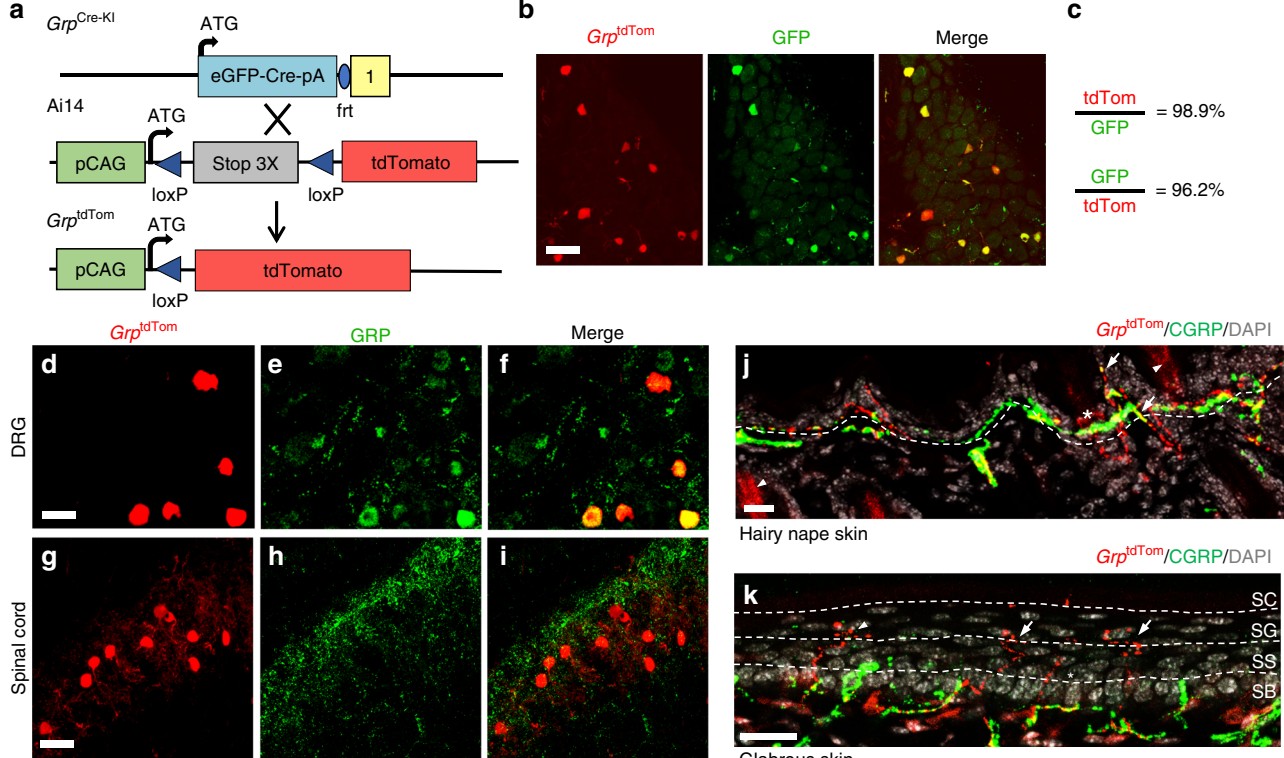

**Fig. 2 Grp^tdTom recapitulates Grp expression in DRG and the spinal cord. a** Schematic of Grp^Cre-KI mating with Ai14 reporter line to produce Grp^tdTom mice. **b** IHC images of eGFP and tdTomato epifluorescence in Grp^tdTom DRG. Scale bar, 50 μm. **c** Percentage of eGFP⁺ neurons co-expressing tdTomato or of tdTomato⁺ neurons co-expressing eGFP. n = 3 mice and 10 sections. **d–f** IHC images of tdTomato epifluorescence (**d**) and GRP (**e**) in Grp^tdTom DRG. Scale bar, 20 μm. **g–i** IHC of tdTomato epifluorescence (**g**) and GRP (**h**) in Grp^tdTom spinal cord. Scale bars, 50 μm. **j** IHC image of tdTomato, CGRP and DAPI in Grp^tdTom nape skin. Arrows indicate apparent Grp^tdTom fibers co-expressing CGRP. Arrowheads indicate autofluorescent hair shafts. Dashed line indicates epidermal/dermal boundary. **k** IHC images of tdTomato, CGRP, and DAPI in glabrous paw skin with stratum basilis (SB), stratum spinosum (SS), stratum granulosum (SG), and stratum corneum (SC) epidermal layers marked by dashed lines. Arrows indicate Grp^tdTom fibers with 'S' or 'Z' pattern nerve endings. Arrowhead indicates Grp^tdTom fibers with apparent bush endings. Scale bars, 20 μm.

**Grp is expressed in a subset of peptidergic sensory neurons.** DRG neurons can be extensively divided into different subsets based upon expression profiles of various molecular markers as well as their size distributions[33,34]. We analyzed Grp^tdTom expression in DRG using several classic neuronal markers. Grp^tdTom sensory neurons co-expressed CGRP (arrowheads) (71.1%, 54/76) or the non-peptidergic Isolectin B4 (IB4)-binding (arrowheads) (26.0%, 33/127) (Fig. 3a–d)[35,36]. Grp^tdTom neurons also co-expressed transient receptor potential cation channel subfamily V member 1 (TRPV1)[37,38] (arrowheads) (74.8%, 86/115) (Fig. 3e, f). Consistent with our finding that Grp^tdTom was not observed in myelinated fibers of skin sensory structures, Grp^tdTom DRG neurons rarely expressed NF-H (2.4%, 5/204) (Supplementary Fig. 3a, b). Next we performed ISH to determine expression of Grp with an itch-specific receptor Mrgpra3[39], or with histamine receptor 1, Hrh1[40]. Mrgpra3 was co-expressed in most Grp neurons (65.2%, 47/72) (Fig. 3e, f), and nearly half of Mrgpra3 neurons co-expressed Grp (41.2%, 47/114). Hrh1 was co-expressed in a majority of Grp neurons (56.2% 36/64 neurons) (Fig. 3g, h), whereas a small portion of Hrh1 neurons co-expressed Grp (20.8%, 36/173). Lastly, we further characterized Grp^tdTom DRG neurons by measuring the area (μm²) of perikarya as well as for CGRP and IB4 neurons (Supplementary Fig. 3c). The areas of Grp^tdTom neuron perikarya were typically less than that of CGRP and IB4 neurons (Grp^tdTom 292 ± 13 μm² vs CGRP 369 ± 16 μm² vs IB4 364 ± 9 μm²). Overall, the size profiles indicate that Grp^tdTom neurons are mostly smaller sized neurons and show similar distributions to CGRP or IB4 populations, albeit shifted to a slightly smaller area profile than CGRP or IB4.

**Grp sensory neurons are activated by pruritogens.** To investigate the responsiveness of Grp sensory neurons to pruritogens, we performed Ca²⁺ imaging of cultured DRG neurons from Grp^tdTom mice. tdTomato⁺ neurons were observed in culture and loaded with the ratiometric indicator fura2-AM for live-cell imaging of intracellular Ca²⁺ levels as described[41](Fig. 4a, b). Chloroquine (CQ) and histamine (Hist), two archetypal pruritogens for non-histaminergic and histaminergic itch, respectively, were applied separately, followed by capsaicin (Cap), a potent agonist of TRPV1[42] which can also signal itch via Mrgpra3 neurons[43], and potassium chloride (KCl) as a positive control response (Fig. 4c, d). We found that ~74% of Grp^tdTom neurons responded to CQ (72/97) and ~52% of CQ-responsive neurons expressed Grp^tdTom (72/138) (Fig. 4e). ~52% Grp^tdTom neurons responded to Hist (51/97) and ~33% of Hist-responsive neurons expressed Grp^tdTom (51/156) (Fig. 4e). Approximately 41% of Grp^tdTom neurons were responsive to both CQ and Hist (40/97) (Fig. 4e). Most Grp^tdTom neurons also responded to Cap (~81%, 79/97) (Fig. 4e). As with DRG tissues, cultured Grp^tdTom DRG neurons co-expressed CGRP, IB4 and TRPV1 but almost no co-expression with NF-H (Supplementary Fig. 4a, b). Taken together, the data suggested that the vast majority of Grp^tdTom sensory neurons are activated by pruritogens.

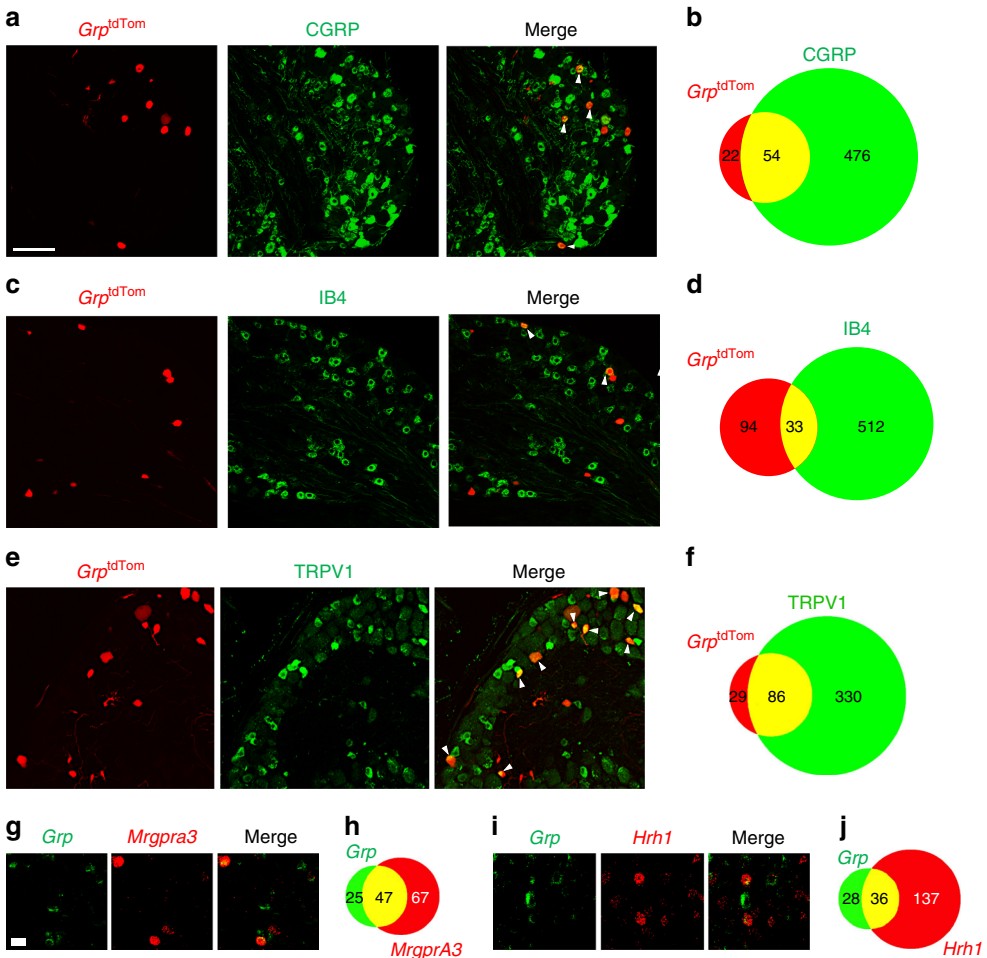

**Fig. 3 *Grp* expression is enriched in a subset of sensory neurons with both peptidergic and non-peptidergic markers. a**, **c**, **e** IHC image of tdTomato with CGRP (**a**), IB4 (**c**), and TRPV1 (**e**) in *Grp*<sup>tdTom</sup> DRG. Arrowheads indicate double stained neurons. Scale bar, 50 μm. **b**, **d**, **f** Venn Diagram of DRG neurons with tdTomato/CGRP expression, tdTomato/IB4-binding, and tdTomato/TRPV1 expression from *Grp*<sup>tdTom</sup> mice. **g** ISH of *Grp* and *Mrgpra3* in WT DRG. Scale bar, 20 μm. **h** Diagram of DRG neurons with *Grp* and *Mrgpra3* expression. **i** ISH of *Grp* and *Hrh1* in WT DRG. **j** Diagram of DRG neurons with *Grp* and *Hrh1* expression. $n = 3$ mice and 9 sections.

**Opto-activation of *Grp* fibers induces itch-specific behavior.** Since *Grp* is also expressed in the spinal cord, it is technically difficult to inactivate *Grp* neurons exclusively in DRGs. To overcome the problem, we crossed *Grp*<sup>Cre-KI</sup> mice with a flox-stop channel rhodopsin-eYFP (ChR2-eYFP) line (Ai32)[44] to generate mice with ChR2-eYFP expression in *Grp* neurons (*Grp*<sup>ChR2</sup>) (Fig. 5a, b) so that *Grp* sensory neurons can be opto-activated to assess their role in itch and pain. As with *Grp*<sup>tdTom</sup>, ChR2-eYFP was detected in both CGRP<sup>+</sup> and IB4-binding sensory neurons in *Grp*<sup>ChR2</sup> mice (Supplementary Fig. 5a), whereas ChR2-eYFP was absent in *Grp*<sup>WT</sup>;Ai32 mice (Supplementary Fig. 5b). eYFP was also detected in nerve fibers of the nape skin and cheek skin that expressed βIII-tubulin (Fig. 5c, Supplementary Fig. 5d), but no ChR2-eYFP was detected in *Grp*<sup>WT</sup>;Ai32 mice (Supplementary Fig. 5c). We also detected *Grp*<sup>ChR2</sup> fibers in the glabrous skin that appeared intertwined or co-expressed with some CGRP<sup>+</sup> fibers in both the dermal and epidermal layers (Supplementary Fig. 5e). Moreover, DiI injection was injected into the nape or cheek skin to retrogradely label DRG neurons[45] that confirmed *Grp*<sup>ChR2</sup> DRG neurons innervate the nape skin (Supplementary Fig. 5f), as well as *Grp*<sup>ChR2</sup> TG neurons that innervated the cheek skin (Supplementary Fig. 5g).

To assess whether activation of *Grp* sensory neurons transmits itch-specific information, we took advantage of *Grp* fiber innervation in the skin and observed corresponding behaviors upon optogenetic activation. *Grp*<sup>ChR2</sup> mice were stimulated with 473 nm blue light with a fiber optic held just above the nape skin (15 mW power from fiber tip) at 1, 5, 10, or 20 Hz with a 3-s On-Off cycle for 5 min total (Fig. 5d). Notably, *Grp*<sup>ChR2</sup> mice exhibited significant scratching behaviors during blue light stimulation trials at 10 and 20 Hz, whereas *Grp*<sup>WT</sup> mice showed almost no scratching (Fig. 5e, f, Supplementary Fig. 6a, Supplementary Movie 1, 2). For the 3 s stimulation trials that induced scratching, the latency to scratch was ~1.7 s (Supplementary Fig. 6b). At 20 Hz, the percentage of trials that induced scratching in *Grp*<sup>ChR2</sup> mice was much greater compared to *Grp*<sup>WT</sup> mice (43.2 vs 2.5%, $p < 0.001$) (Supplementary Fig. 6c). Similarly, the total number of scratches induced during the 5 min test at 20 Hz was significant compared to *Grp*<sup>WT</sup> mice (24.3 vs 1.9 scratches, $p < 0.001$) (Fig. 5g). To determine whether the blue light-induced scratching behaviors were indicative of pain or itch, we treated *Grp*<sup>ChR2</sup> mice with systemic morphine (5 mg kg<sup>−1</sup> intraperitoneal) for analgesia since morphine-induced scratching and analgesia are mediated by distinct pathways in the spinal cord[46]. We found that morphine analgesia in *Grp*<sup>ChR2</sup> mice had no significant effect on scratching induced by 20 Hz blue-light stimulation (Fig. 5g) or on the percentage of trials inducing scratching behavior (Supplementary Fig. 6c). This

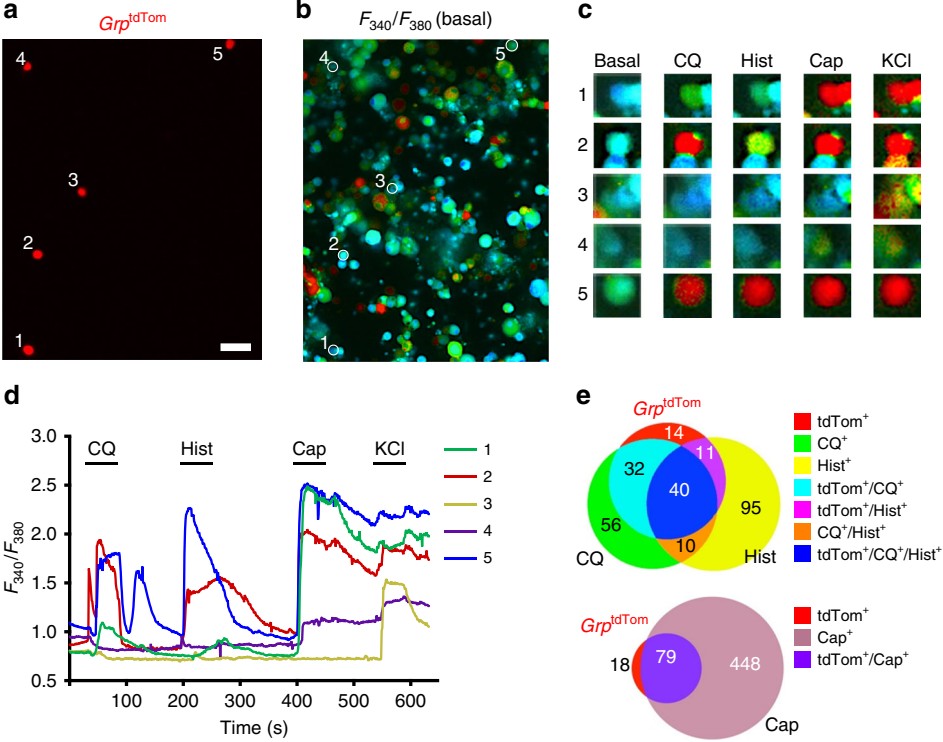

**Fig. 4 Pruritogens induce Ca²⁺ responses in *Grp*⁺ sensory neurons. a** and **b** Image of tdTomato neurons (**a**) and $F_{340}$ / $F_{380}$ signal (**b**) from *Grp*^tdTom DRG cultures loaded with fura 2-AM. Scale bar, 50 µm. **c** Snapshots of *Grp*^tdTom neuron intracellular Ca²⁺ levels at basal and 20 s after CQ (1 mM), Hist (100 µM), Cap (100 nM) or KCl (30 mM) applications. **d** $F_{340}/F_{380}$ traces from individual *Grp*^tdTom neurons with application of CQ, Hist, Cap and KCl. **e** Venn diagram of *Grp*^tdTom DRG neurons responsive to CQ, Hist or Cap and/or expressing tdTomato. $n$ = 4 mice and 1424 neurons for **a**–**e**. Source data are provided as a Source Data file.

finding suggested that scratching is likely to be itch-related rather than pain. To confirm this, intrathecal injection (i.t.) of bombesin-saporin (BB-sap, 200 ng) was performed to ablate GRPR neurons in the spinal cord and block itch transmission[47]. Indeed, *Grp*^ChR2 mice treated with BB-sap showed a significant reduction in scratching (24 vs 8 scratches, $p < 0.001$) (Fig. 5g) as well as the percentage of trials inducing scratching behaviors at 20 Hz (43.2 vs 14.4%, $p < 0.001$) (Supplementary Fig. 6c). We monitored spontaneous scratching behaviors for 30 min prior to and 30 min following the 5 min-20 Hz stimulation test and found a significant increase in scratching following stimulation in *Grp*^ChR2 mice compared to *Grp*^WT (68 vs 14 scratches, $p < 0.001$), which was not affected by morphine analgesia but was almost completely blocked in BB-sap treated mice (68.4 vs 10.1, $p < 0.001$)(Supplementary Fig. 6d). Lastly, we crossed *Grp*^Cre-KI mice with a Flox-stop hM3Dq DREADD (designer receptors exclusively activated by designer drugs) line (R26-LSL-Gq-DREADD)[48] to generate mice with hM3Dq expression in *Grp* neurons (*Grp*^Gq), so that *Grp* neurons can be activated by clozapine-N-oxide (CNO). To assess the role of *Grp* sensory neurons in itch and pain, we employed the cheek model that can differentiate itch behaviors as scratching with the hind limb and pain behaviors as wiping with the forelimb[49]. Notably, we found that injection of CNO into the cheek skin of *Grp*^Gq mice induced robust scratching responses (36 scratches in 30 min) (Supplementary Fig. 6e). In contrast, *Grp*^WT mice barely showed cheek scratching responses to CNO injection (4 scratches in 30 min) due to lack of DREADD (Supplementary Fig. 6e). Importantly, both *Grp*^Gq and *Grp*^WT mice showed minimal wiping behaviors to CNO injections into the cheek (Supplementary Fig. 6e). These data indicate that activation of *Grp* sensory neurons selectively transmits itch sensation.

**Conditional deletion of sensory *Grp* attenuates itch behavior.** Previous evidence from global *Grp* KO mice indicated that loss of GRP attenuated non-histaminergic itch behavior, whereas histaminergic itch and pain transmission were normal[16]. However, it remained unclear whether the source of GRP from sensory neurons, spinal neurons or the brain were important for itch. To determine the role of GRP in sensory neurons, we generated a floxed *Grp* allele and crossed it with the sensory neuron specific Cre-recombinase line, Na$_V$1.8^Cre [50], to conditionally delete *Grp* expression in sensory neurons (Fig. 6a). *Grp* mRNA expression was mostly absent in DRG of *Grp*^F/F;Na$_V$1.8^Cre mice compared to *Grp*^F/F control littermates (Fig. 6b), but *Grp* mRNA expression in the dorsal horn was not affected (Supplementary Fig. 7a). Consistent with ISH data, IHC of GRP showed that GRP⁺ neurons were rarely seen in DRG of *Grp*^F/F;Na$_V$1.8^Cre mice, demonstrating the deletion of GRP in DRGs (Fig. 6c). As predicted, GRP⁺ primary afferents in the dorsal spinal cord of *Grp*^F/F;Na$_V$1.8^Cre mice were also mostly absent (Fig. 6d, e). Western blot analysis using the same GRP antibody detected a 15.6-kDa band that corresponds to the size of GRP preproprotein in *Grp*^F/F DRG and the spinal cord (Fig. 6f, lane 1, red arrow), but only a faint band in DRG of *Grp*^F/F;Na$_V$1.8^Cre mice, with a normal non-specific band of 30 kDa as an internal control (Fig. 6f, lane 2). Importantly, the GRP band in the spinal cord of *Grp*^F/F;Na$_V$1.8^Cre mice was also significantly reduced compared to the control (Fig. 6f, lane 2). These results indicate that the vast majority of GRP protein in the spinal cord is of primary afferent terminal origin, consistent with our previous results[19]. Notably, GRP IHC in the SCN is comparable between *Grp*^F/F and *Grp*^F/F;Na$_V$1.8^Cre littermates, as we previously described[10], further demonstrating the specificity of GRP deletion in sensory neurons (Supplementary Fig. 7b). Next, we performed acute itch behavior tests and found significant

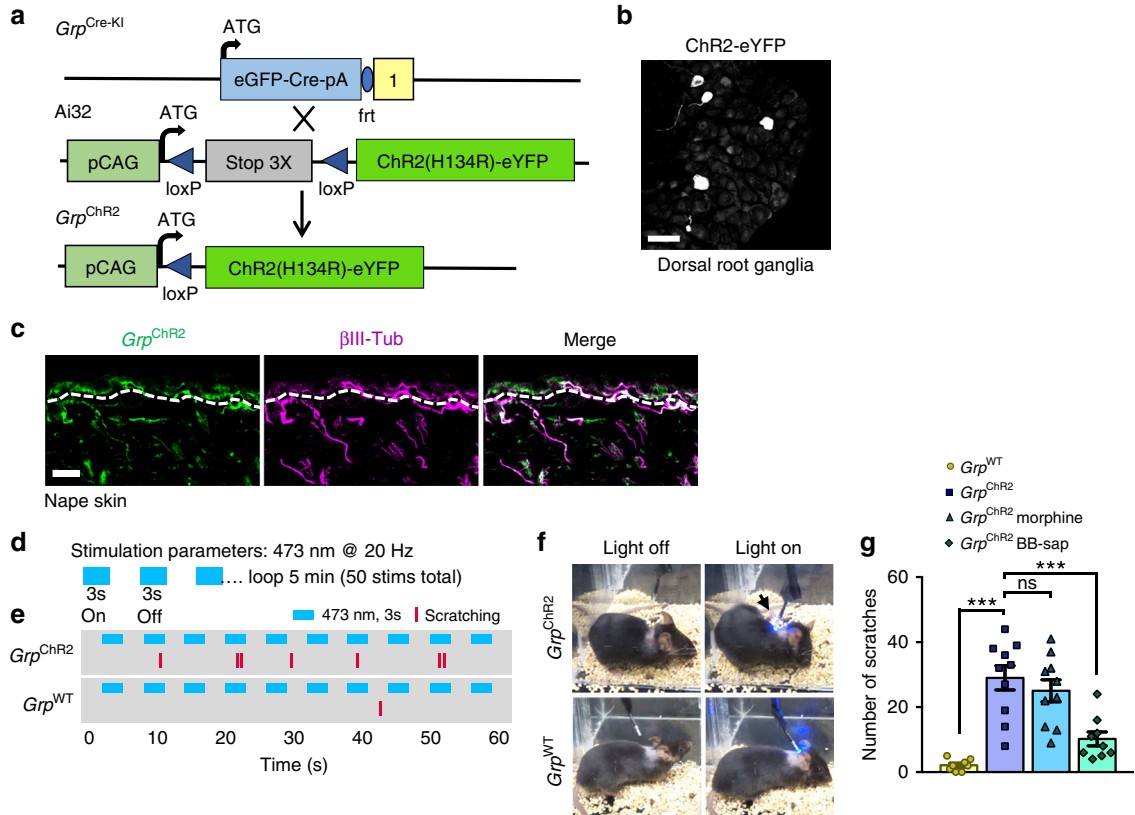

**Fig. 5 Opto-activation of *Grp*+ sensory neuron skin fibers evokes itch behavior. a** Schematic of *Grp*^Cre-KI mating with Ai32 ChR2-eYFP line to produce *Grp*^ChR2 mice. **b** IHC Image of eYFP expression in *Grp*^ChR2 DRG. Scale bar, 50 μm. **c** IHC images of eYFP and βIII-Tubulin in *Grp*^ChR2 nape skin. Dashed lines indicate epidermal/dermal boundary. Scale bar, 100 μm. **d** Optical parameters of skin fiber stimulation *Grp*^ChR2 and *Grp*^WT mice. **e** Raster plot of scratching behavior induced by light stimulation of skin in *Grp*^ChR2 and *Grp*^WT mice. **f** Snapshots of *Grp*^ChR2 and *Grp*^WT mice with light off or on. Arrow indicates hind paw scratching the nape when light is on. **g** Total number of scratches during 5-min light stimulation experiment in *Grp*^WT, *Grp*^ChR2, *Grp*^ChR2 morphine-treated and *Grp*^ChR2 BB-sap-treated mice. Data are presented as mean ± s.e.m., $n = 8$ mice for *Grp*^WT, 9 mice for *Grp*^ChR2 BB-sap, and 10 mice for *Grp*^ChR2 and *Grp*^ChR2 morphine, one-way ANOVA with Tukey post hoc, ***$p < 0.001$, ns not significant. Source data are provided as a Source Data file.

reductions of scratching behavior in *Grp*^F/F;Na_V1.8^Cre mice compared to *Grp*^F/F control for the non-histaminergic pruritogens CQ ($240 \pm 24$ vs $143 \pm 16$ scratches, $p < 0.01$) (Fig. 6g), the PAR2 agonist SLIGRL-NH2[51] ($222 \pm 17$ vs $109 \pm 23$ scratches, $p < 0.01$) (Fig. 6h), and the MrgprC11 agonist BAM8-22[52] ($46 \pm 4$ vs $29 \pm 3$ scratches, $p < 0.01$) (Fig. 6i). However, *Grp*^F/F;Na_V1.8^Cre mice and *Grp*^F/F control showed similar scratching behavior for histamine ($53 \pm 6$ vs $51 \pm 7$ scratches, $p = 0.79$) (Fig. 6j) and β-alanine ($36 \pm 27$ vs $52 \pm 29$ scratches, $p = 0.31$) (Fig. 6k), the latter of which is mediated by MrgprD[53]. Recently, Albisetti et al.[24] reported that *Grp* neurons in the lumbar spinal cord is involved in pruritogens-evoked biting behaviors to the calf. The biting behavior in the calf model has been used as a readout for itch[54,55]. To validate the model, we examined biting/licking time after calf injections of pruritogens and found that *Grp*^F/F;Na_V1.8^Cre mice and *Grp*^F/F control showed similar responses to CQ ($98 \pm 17$ vs $107 \pm 28$ s, $p = 0.78$), SLIGRL-NH2 ($114 \pm 30$ vs $88 \pm 24$ s, $p = 0.52$), and BAM8-22 ($82 \pm 9$ vs $78 \pm 10$ s, $p = 0.76$) (Supplementary Fig. 7c–e). We noticed that mice showed robust responses to saline upon calf injections ($49 \pm 8$ s)(Supplementary Fig. 7f), suggesting that the injection itself caused irritation to the animals. To evaluate if the biting/licking behaviors reflect itch, we performed i.t. injection of morphine (0.3 nmol, 30 min) and observed its effect on calf CQ injection. We found that morphine abolished biting/licking responses to calf CQ injection compared to saline ($116 \pm 16$ vs $10 \pm 3$ s, $p < 0.001$) (Supplementary Fig. 7g). CQ-induced itch is dependent on GRPR neurons, which can be

ablated by i.t. bombesin-saporin (BB-sap)[47]. We found that calf CQ injection-evoked biting/licking behaviors were not affected in BB-sap-treated mice (Supplementary Fig. 7h). These results indicate that injection of a pruritogen into the calf skin more likely evokes inflammatory pain rather than itch responses in mice.

**Spinal *Grp* neurons are dispensable for itch and pain.** To test the role of spinal *Grp* interneurons in transmitting itch and pain behaviors while not affecting *Grp* expressing neurons in the DRG and brain, we used an intersectional genetic approach[56,57] to ablate *Grp* neurons specifically in the dorsal spinal cord and spinal trigeminal nucleus subcaudalis (SpVc). *Grp*^Cre-KI mice were crossed with the *Lbx1*^Flpo and *Tau*^ds-DTR lines to express the diphtheria toxin receptor (DTR) exclusively in *Grp* spinal neurons and mice were injected with diphtheria toxin (DTX) to specifically ablate *Grp* spinal neurons (Fig. 7a). We tested the efficiency of DTR-mediated ablation in the spinal cord by *Grp* ISH (Fig. 7b–e) as well as *Grp*^tdTom neurons (Fig. 7f–i). Compared to control littermates, DTX injection resulted in ablation of ~82% of *Grp* neurons and ~93% of *Grp*^tdTom dorsal horn neurons in ablated mice (Fig. 7j). Importantly, we confirmed that *Grp*^tdTom neurons in the DRG and brain (cingulate cortex, hippocampus) were not ablated following DTX injection (Supplementary Fig. 8a). We analyzed spinal interneuron populations expressing PKCγ, Pax2, and *Grpr*, and found similar numbers of

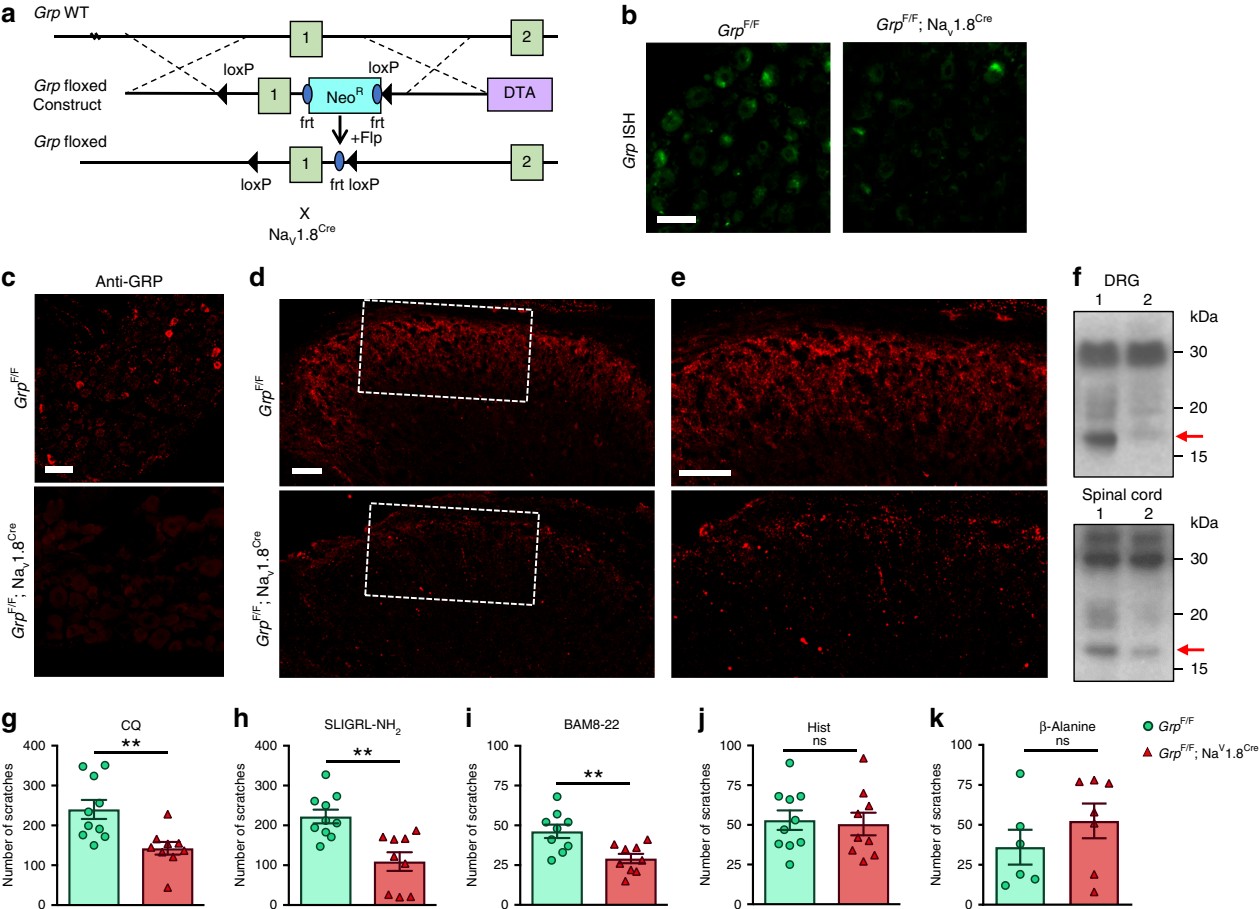

**Fig. 6 Attenuated itch behaviors in mice with conditional deletion of *Grp* in sensory neurons. a** Schematic of targeting strategy for inserting loxP sites into *Grp* allele to generate *Grp* floxed (*Grp*$^{F/F}$) mice and subsequent crossing with Na$_v$1.8$^{Cre}$ mice. **b** ISH images of *Grp* expression in DRG sections from *Grp*$^{F/F}$ and *Grp*$^{F/F}$; Na$_v$1.8$^{Cre}$ mice. Scale bar, 50 μm. **c** IHC images of GRP antibody in DRG sections from *Grp*$^{F/F}$ and *Grp*$^{F/F}$; Na$_v$1.8$^{Cre}$ mice. Scale bar, 50 μm. **d** IHC images of GRP antibody in spinal cord sections from *Grp*$^{F/F}$ and *Grp*$^{F/F}$; Na$_v$1.8$^{Cre}$ mice. Scale bar, 100 μm. **e** High power images of boxed area in **d**. Scale bar, 100 μm. **f** Western blot of GRP in DRG (upper row) and the spinal cord (lower row) from *Grp*$^{F/F}$ (lane 1) and *Grp*$^{F/F}$; Na$_v$1.8$^{Cre}$ mice (lane 2). Arrow, predicted size of pro-GRP at 15.6 kDa. **g–k** Mean number of scratches induced by i.d. nape injection of (**g**) CQ (200 μg), (**h**) SLIGRL-NH$_2$ (100 μg), (**i**) BAM8-22 (100 μg), (**j**) Hist (200 μg), and (**k**) β-alanine (100 μg) in *Grp*$^{F/F}$ and *Grp*$^{F/F}$; Na$_v$1.8$^{Cre}$ littermates. Data are presented as mean ± s.e.m., $n = 9$ mice for *Grp*$^{F/F}$ and 10 mice for *Grp*$^{F/F}$;Na$_v$1.8Cre in **g–j**. $n = 6$ mice for *Grp*$^{F/F}$ and 7 mice for *Grp*$^{F/F}$;Na$_v$1.8Cre in **k**. **$p < 0.01$, unpaired two tails $t$ test in **g–k**. Source data are provided as a Source Data file.

neurons in control and ablated mice (Supplementary Fig. 8b, c). *Npr1* neurons were reduced slightly but did not reach statistical significance ($77 \pm 3$ vs $68 \pm 3$ neurons, $p = 0.07$) (Supplementary Fig. 8b, c). This was supported by single-cell RT-PCR results showing 20–25% co-expression of *Npr1* or *Npr3* in *Grp*$^{tdTom}$ neurons (Supplementary Fig. 9) as well as no co-expression of *Grp*$^{tdTom}$ in PKCγ cell bodies (Supplementary Fig. 8b). The small percentage of overlap between spinal *Grp* and *Npr1* is consistent with recent studies using single nucleus RNA sequencing (snRNA-seq) of spinal cord neurons[58]. Moreover, RNAscope ISH showed that ~40% of NMBR neurons express *Grp*, whereas about 25% of *Grp* neurons express TacR3, a receptor implicated in pain[59] (Supplementary Fig. 9). To determine the role of spinal *Grp* neurons in itch and pain transmission, we tested acute itch and pain behaviors pre- and post-DTX injection in control and ablated mice. Scratching behaviors for CQ and histamine were similar prior to and following DTX injection in control and ablated mice with no significant differences (Fig. 7k, l). Our finding is in accordance with the observation that few *Grp* neurons seldom expressed c-Fos in response to CQ injection[60]. Wiping behaviors for capsaicin cheek injection were also comparable between control and ablated mice both pre- and

post-DTX injection (Fig. 7m). Thermal response latencies for hot plate and Hargreaves hind-paw assays were comparable in control and ablated mice both pre- and post-DTX injection, as well as no changes in mechanical thresholds by von Frey filament assay (Fig. 7n–p). Taken together, the finding that specific ablation of spinal *Grp* neurons does not significantly affect itch or pain behavior responses suggests that spinal *Grp* neurons are dispensable for itch and pain transmission.

## Discussion

Using newly generated floxed *Grp* mice, we demonstrate the role of GRP in sensory neurons for CQ- but not histamine-evoked itch behavior, in agreement with the phenotype of global *Grp* KO mice[5,16,47]. Analysis of the *Grp*$^{Cre-KI}$ mouse line reveals a widespread innervation of the skin by GRP fibers whose activation by optogenetics and chemogenetics evokes itch-related scratching behavior. Importantly, intersectional genetic ablation approach demonstrates that spinal *Grp* neurons are dispensable for itch transmission.

Despite co-expression of CGRP in the cell bodies of some *Grp* neurons, *Grp* epidermal fibers mostly lack detectable levels of

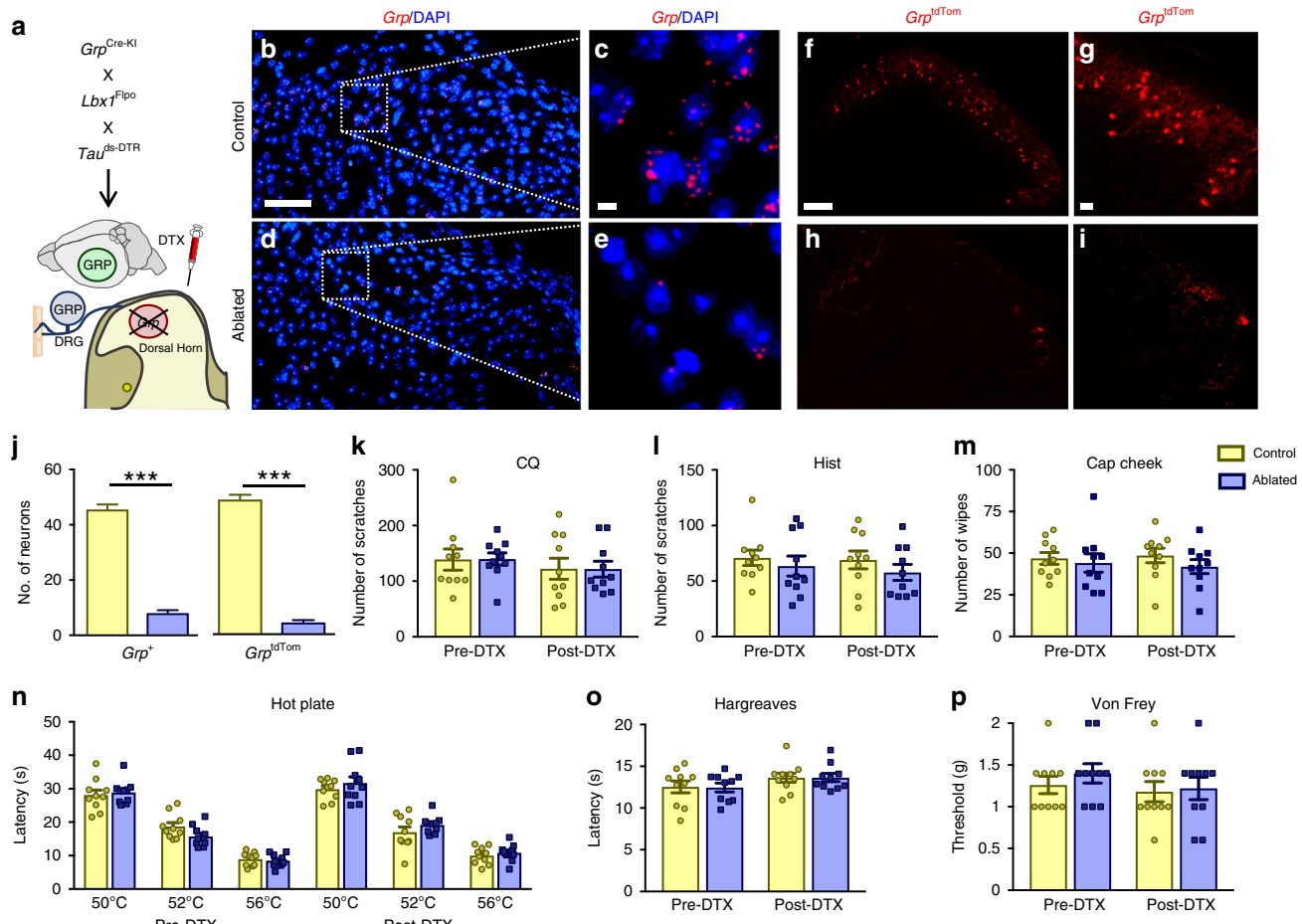

**Fig. 7 Intersectional ablation of *Grp* spinal neurons does not affect itch or pain behavior responses. a** Schematic of mating strategy of *Grp*^Cre-KI with *Lbx1*^Flpo and *Tau*^ds-DTR lines to generate DTR-expressing *Grp* spinal neurons and subsequent DTX injection to ablate *Grp* spinal neurons. **b–e** ISH images of *Grp* expression in cervical spinal cord sections from (**b, c**) control and (**d, e**) ablated littermates. Scale bars, 100 μm in **b** and 20 μm in **c**. **f–i** Epifluorescent images of tdTomato in cervical spinal cord sections from (**f, g**) *Grp*^tdTom control and (**h, i**) *Grp*^tdTom ablated mice. Scale bars, 100 μm in **f** and 20 μm in **g**. **j** Mean number of *Grp* + neurons and *Grp*^tdTom neurons in control and ablated sections. n = 3 mice and 10 sections, unpaired *t* test, ***p < 0.001. **k, l** Mean number of scratches induced by i.d. nape injection of (**k**) CQ (100 μg) and (**l**) Hist (200 μg) in control and ablated mice pre- and post-DTX injection. **m** Mean number of wipes induced by i.d. cheek injection of Cap (20 μg) in control and ablated mice pre- and post-DTX injection. n Mean response latencies for hot plate assay at 50 °C, 52 °C, and 56 °C in control and ablated mice pre- and post-DTX injection. **o** Mean response latencies for Hargreaves hind-paw assay in control and ablated mice pre- and post-DTX injection. **p** Withdrawal thresholds (in grams of force, g) for Von Frey filament assay in control and ablated mice pre- and post-DTX injection. n = 10 mice, two-way RM ANOVA with Tukey post-hoc in **k–p**. Data are presented as mean ± s.e.m. Source data are provided as a Source Data file.

CGRP, indicating differences in expression/trafficking of CGRP between cell body and free nerve endings. Moreover, the morphological characteristics of *Grp* epidermal fibers and their elaborate innervation patterns in the stratum granulosum are consistent with a non-peptidergic profile that was previously described in *Mrgprd* fibers[30]. That a small percentage of *Grp* neurons gave rise to extensive branching in the skin is in sharp contrast to a progressively restriction of CGRP expression in the terminals, possibly reflecting the difference in terminal transportation of neuropeptides. The broad innervation of *Grp* fibers matches dense innervation of the superficial dorsal horn by central terminals of GRP primary afferents. The finding that more than 60% of *Mrgpra3* express GRP is in support of previous studies indicating that *Mrgpra3* afferents form synaptic contacts with GRPR neurons and use GRP to transmit CQ as well as other non-histaminergic itch[39,43,47]. In line with this, GRP and/or *Mrgpra3* were found to be up-regulated in DRGs of mice with chronic itch[7,19,21,61]. Of note, not all *Grp* neurons were responsive to CQ, histamine, and capsaicin, as measured by calcium

imaging studies. These non-responding neurons may express receptors for pruritogens which were not investigated in our study. Given increasing numbers of receptors/TRP channels in DRGs have been implicated in itch[62–66], it is likely that modality specificity is in part encoded through the differential activation and release of myriad neuropeptides. Whether a broad innervation of the epidermis by a small subset of pruriceptors may reflect a signature of modality-specific sensory neurons remains to be determined.

Of several putative itch-related primary afferents, to date only scratching behavior evoked by optogenetic and chemogenetic activation of cutaneous GRP fibers has been validated to be itch-related. The reduced GRP IHC staining in the central terminals after CQ injection provides a causal link between itch-related scratching behavior and potential release of GRP into the spinal cord. Notably, in spite of overlapping expression of SP and GRP in DRGs[19], no reduction of SP IHC was observed in response to CQ, raising the possibility that GRP and SP are differentially released onto the spinal cord, depending on the nature of stimuli.

The fact that mice spend significant amount of awake time scratching themselves implies a constant spontaneous release of GRP. This contrasts the difficulties of evoked SP release, which likely induces pain-like behavior. Hence, distinct mechanisms may control release of GRP vs. SP from the large dense core vesicles in the central terminals, respectively. To the best of our knowledge, our study represents the first comprehensive validation of itch-related scratching behavior evoked by cutaneous activation of a subset of primary afferents using a combination of approaches, including both nape and cheek models. Given the complexity of scratching behaviors in mice, our strategy as described here can be used to differentiate between itch- and pain-related scratching behavior and thus to identify itch- or pain-related primary afferents in sensory neurons. The present study also cautions the use of alternative behavioral outputs (e.g. licking/biting) as a proxy for itch behavior.

Prior studies with global ablation of *Grp*-Cre neurons found impaired itch and enhanced pain behaviors[25]. The deficits are more likely caused by ablation of *Grp*-Cre neurons in the brain rather than the spinal cord since *Grp*-Cre neurons are found in several pain-modulating areas in the brain, including mPFC or ACC. Moreover, while spinal *Grp* neurons do not express PKCγ [67], Sun et al. ablated ~50% of PKCγ neurons, indicating extensive ectopic ablation even in the dorsal horn neurons[25]. By contrast, our intersectional strategy did not impact PKCγ expression. Based on the premise that GRP is not expressed in DRGs, a recent study suggested that repeated stimulation of spinal *Grp*-Cre neurons activated GRPR neurons through GRP/glutamate released from spinal *Grp* neurons[24]. However, the release of GRP can be alternatively attributable to concurrent stimulation of *Mrgpra3*/GRP afferents which innervate spinal *Grp* neurons[23]. Furthermore, given *Mrgpra3* fibers also express NMB[68], which can activate *Grp*/NMBR neurons, conceivably, stimulation of *Grp*/NMBR neurons per se, either directly, or indirectly, could activate downstream GRPR neurons via glutamatergic transmission as we previously described[16]. It should be noted that these possibilities are not mutually exclusive. Importantly, the observation of normal itch and pain behavior of mice with spinal ablation of *Grp* neurons should not be mistaken for their unimportance. On the contrary, *Grp* neurons could be an integral part of distinct "gates" or "labeled lines" for itch and pain transmission, and the net effect of deletion of a small percentage of which conceivably could be negligible at behavioral level.

Our finding that *Grp* neurons are molecularly and functionally heterogenous raises an important question of what may constitute the appropriate criteria for categorizing the dorsal horn interneurons concerning the coding of sensory modality. With the aid of powerful chemogenetic/optogenetic/ablation approaches, recent studies have capitalized on a dozen of Cre lines, mostly based on neuropeptides, transcription factors, or generic enzymes/channels/transporters, in identification of a number of "gates" for itch and pain. Such circuit-based approaches, albeit necessary, might be insufficient, or in some cases could even complicate our ability to identify the true labeled lines in the spinal cord if the Cre expression is either widespread in the dorsal horn or cross several discrete microcircuits. Given dozens of neuropeptides identified in sensory neurons[69], predominant role of GPCRs in pain and itch[70], and the exquisite modular organization of laminae I–II circuitry that permits dedicated transmission of sensory information[71], an understanding of how information encoded by neuropeptides in DRGs is transmitted and processed by their respective GPCRs in laminae I–II could be essential for decoding sensory processing. In this regard, a GPCR-based Cre line could be advantageous since it affords functional, electrophysiological, and pharmacological interrogation of microcircuits in the context of input–output flow. After all,

our goal is to understand how discrete somatosensory information is transmitted from sensory neurons to the spinal cord. While expression of GPCRs may overlap, by examining ligand-mediated activation under both basal and various physiological conditions, one may still be able to infer the coding logic by which distinct modality-specific microcircuits are differentially modulated.

In conclusion, using conditional deletion of GRP in sensory neurons, we have established the role of GRP as an itch-specific neuropeptide in transmitting nonhistaminergic itch. Using intersectional genetic ablation of spinal *Grp*^Cre-KI neurons, we have shown that spinal *Grp* neurons are dispensable for itch and pain transmission. By using the gold-standard behavioral assay for itch, our data strongly suggest that GRP primary afferents constitute a labeled line for itch. The newly generated mouse lines, along with floxed *Grpr*[10], *Grpr*^iCre [72] we previously generated, offer unprecedented access to the GRP-GRPR system for unraveling the coding logic of itch as well as the phenotypic change consequent to the development of chronic itch in a cell-type- and region-specific manner.

## Methods

**Mice**. Mice between 7 and 12 weeks old were used for experiments. Ai14 (Stock no. 007908), Ai32 (Stock no. 024109), R26-LSL-Gq-DREADD (Stock no. 026220), and C57Bl/6J mice were purchased from Jackson Laboratory, the Na$_v$1.8^Cre line[50] was kindly provided by the J. Wood lab, and the *Lbx1*^Flpo and *Tau*^ds-DTR lines[56,57] were kindly provided by the Q. Ma and M. Goulding labs. Mice were housed in clear plastic cages with no more than 5 mice per cage in a controlled environment at a constant temperature of ~23 °C and humidity of 50 ± 10% with food and water available ad libitum. The animal room was on a 12/12-h light/dark cycle with lights on at 7 am. All experiments were performed in accordance with the guidelines of the National Institutes of Health and the International Association for the Study of Pain and were approved by the Animal Studies Committee at Washington University School of Medicine.

**Generation of *Grp*^Cre-KI mice**. A ~6-kb genomic DNA fragment containing *Grp* exon 1 was PCR amplified from a BAC clone (RP-23, id. 463E10, ThermoFisher Scientific) with a high-fidelity DNA polymerase (CloneAmp HiFi PCR mix, Clontech) and subcloned into pBluescript KS. An eGFP-Cre and frt-flanked PGK-Neomycin cassette were integrated into exon 1 ATG start site using an In-Fusion® HD Cloning Kit (Clontech). A diphtheria toxin α (DTA) cassette was inserted downstream of the 3′ homology arm of the construct as a negative selection marker. The construct was linearized with AscI and electro-rporated into AB1 embryonic stem cells. Following negative selection with G418 (200 ng μL$^{-1}$), positive clones were identified by 5′ and 3′ PCR screening. Positive clones were injected into C57Bl/6J blastocysts to generate chimeric mice. After breeding, germ-line transmission was confirmed by genotype PCR, mice were bred to a FRT-deleter line (Stock. No. 007844, Jackson Laboratory) to remove the Neomycin cassette and were subsequently backcrossed into the C57Bl/6J background.

**Generation of *Grp* floxed mice**. A ~10-kb genomic DNA fragment containing *Grp* exon 1 was amplified and loxP sites were integrated into the fragment to flox exon 1. A frt-flanked PGK-Neomycin cassette for positive selection was integrated into intron 1. A diphtheria toxin α (DTA) cassette was inserted downstream of the 3′ homology arm of the construct as a negative selection marker. The construct was linearized with KpnI and electroporated into AB1 embryonic stem cells. Following negative selection with G418 (200 ng μL$^{-1}$), positive clones were identified by 5′ and 3′ PCR screening. Positive clones were injected into C57Bl/6J blastocysts to generate chimeric mice. After breeding, germ-line transmission was confirmed by genotype PCR, mice were bred to an FRT-deleter line (Stock. No. 007844, Jackson Laboratory) to remove the Neomycin cassette and were subsequently bred with Na$_v$1.8^Cre mice or backcrossed into the C57Bl/6J background.

**In situ hybridization**. Conventional ISH was performed using a digoxigenin-labeled cRNA (Roche) antisense probe for *Grp* (bases 149–707 of *Grp* mRNA, NCBI accession NM_175012.4)[19]. In all, 7–12-week-old mice were anesthetized (ketamine, 100 mg kg$^{-1}$ and Xylazine, 15 mg kg$^{-1}$) and perfused intracardially with DEPC-treated PBS pH 7.4 followed by 4% paraformaldehyde (PFA) in DEPC-treated PBS. Tissues were dissected, post-fixed for 2–4 h, and cryoprotected in 20% sucrose in PBS overnight at 4 °C. Cervical, thoracic and lumbar DRGs were sectioned in OCT at 20 μm thickness at 20 °C using a cryostat microtome (Leica Instruments) and thaw-mounted onto Super Frost Plus slides (Fisher). Slides were incubated with Proteinase K (50 μg mL$^{-1}$) buffer for 10 min, incubated in prehybridization solution for 3 h at 65 °C and then incubated with *Grp* probe (2 μg mL$^{-1}$) in hybridization solution overnight at 65 °C.

After SSC stringency washes and RNase A (0.1 µg mL$^{-1}$ for 30 min) incubation, sections were incubated in 0.01 M PBS with 20% sheep serum and 0.1% Tween blocking solution for 3 h and then incubated with anti-digoxigenin antibody conjugated to alkaline phosphatase (0.5 µg mL$^{-1}$, Roche) in blocking solution overnight at 4 °C. After washing in PBS with 0.1% Tween, sections were incubated in NBT/BCIP substrate solution at room temperature for 12–16 h for colorimetric detection. Reactions were stopped by washing in 0.5% paraformaldehyde in PBS. Bright field images were taken using a Nikon Eclipse Ti-U microscope with a Nikon DS-Fi2 Camera.

RNAScope™ Double ISH (Advanced Cell Diagnostics, Inc.) was performed using the RNAScope™ Multiplex Fluorescent Reagent Kit v2 User Manual for Fixed Frozen Tissue[27]. Tissues were dissected, post-fixed for 2–4 h, and cryoprotected in 20% sucrose in PBS overnight at 4 °C. Cervical, thoracic and lumbar DRGs or cervical spinal cord regions were sectioned in OCT at 20 µm thickness at 20 °C using a cryostat microtome (Leica Instruments) and thaw-mounted onto Super Frost Plus slides (Fisher). Briefly, slides were incubated with hydrogen peroxide for 10 min, washed, mildly boiled in target retrieval reagents for 15 min, washed, dried, and hydrophobic barriers were added around the sections. Protease III Plus Reagent was applied for 15 min, washed and sections were incubated with target probes for Grp (Mm-Grp-C1, GenBank: NM_175012.4, target region bases 22–825, Cat No. 317861), Cre (Cre-O1-C3, GenBank: NC_005856.1, target region bases 2–972, Cat No. 474001-C3), Grpr (Mm-Grpr-C1, GenBank: NM_008177.2, target region bases 463–1596, Cat No. 317871) or Npr1 (Mm-Npr1-C1, GenBank: NM_008727.5, target region bases 941–1882, Cat No. 484531) probes for 2 h. All target probes consisted of 20 ZZ oligonucleotides and were obtained from Advanced Cell Diagnostics. Following probe hybridization, sections underwent a series of probe signal amplification steps followed by incubation of fluorescently labeled probes designed to target the specified channel associated with each probe. Slides were counterstained with DAPI, and coverslips were mounted with FluoromountG (Southern Biotech). Images were taken using a Nikon C2 + confocal microscope system (Nikon Instruments, Inc.) and analysis of images was performed using ImageJ software from NIH Image (version 1.34e)[73]. Positive signals were identified as punctate dots and clusters present around the nucleus and/or cytoplasm. For Grp/Cre mRNA co-expression, dot clusters of C1 and C3 channels associated within a DRG cell body were counted as double positive, whereas neurons with only C1 or C3 dot clusters were counted as single positive.

**Immunohistochemistry**. Mice were anesthetized (ketamine, 100 mg kg$^{-1}$ and Xylazine, 15 mg kg$^{-1}$) and perfused intracardially with PBS pH 7.4 followed by 4% paraformaldehyde (PFA) in PBS[19]. Tissues were dissected, post-fixed for 2–4 h, and cryoprotected in 20% sucrose in PBS overnight at 4 °C. For glabrous and mystacial skin, tissues from 3-week-old mice were harvested immediately following anesthesia overdose, fixed in 2% PFA-PBS with 30% (v/v) Picric acid overnight at 4 °C and cryoprotected in 20% sucrose in PBS. Tissues were sectioned in OCT at 20 µm thickness for DRG and spinal cord and 30 µm for all other tissues including tongue, cornea, bladder, muscle, intestine, esophagus, heart, testis, liver, lung, kidney, and skin. Free-floating frozen sections were blocked in a 0.01 M PBS solution containing 2% donkey serum and 0.3% Triton X-100 followed by incubation with primary antibodies overnight at 4 °C, washed three times with PBS, secondary antibodies for 2 h at room temperature and washed again three times. Sections were mounted on slides and ~100 µL FluoromountG (Southern Biotech) was placed on the slide with a coverslip. The following primary antibodies were used: rabbit anti-GFP (1:1000, Molecular Probes, A11122), chicken anti-GFP (1:500, Aves Labs, GFP-1020), rabbit anti-GRP (1:500, Immunostar, 20073), guinea pig anti-SP (1:1000, Abcam, ab10353), rabbit anti-CGRP (1:3000, MilliporeSigma, AB15360), guinea-pig anti-TRPV1 (1:800, Neuromics, GP14100), chicken anti-NF-H (1:2000, EnCor Biotechnology, CPCA-NF-H), rabbit anti-βIII-Tubulin (1:2000, Biolegend, 802001), rabbit anti-PKCγ (1:1000, Santa Cruz Biotechnology, SC-211), rabbit anti-Pax2 (1:300, ThermoFisher Scientific, 71–6000) IB4-binding was performed using an IB4-AlexaFluor 488 conjugate (2 µg mL$^{-1}$, ThermoFisher Scientific, I32450) and DAPI (2 µg mL$^{-1}$, Sigma, D9542) was used as a nuclear stain of cells. The following secondary antibodies were used: Alexa-Fluor 488 conjugated donkey anti-rabbit (1:1000, Jackson Immuno-Research, 711–545–152), Alexa-Fluor 488 conjugated donkey anti-chicken (1:1000, Jackson ImmunoResearch, 703–545–155), Alexa-Fluor 488 conjugated donkey anti-guinea pig (1:1000, Jackson ImmunoResearch, 706–545–148), Cy3-conjugated donkey anti-rabbit (1:1000, Jackson ImmunoResearch, 711–165–152) and Cy5-conjugated donkey anti-chicken 1:1000, Jackson ImmunoResearch, 703–175–155). Fluorescent Images were taken using a Nikon C2+ confocal microscope system (Nikon Instruments, Inc.) and analysis of images for neuron counting was performed using ImageJ software from NIH Image (version 1.34e).

**Virus injection into DRG**. AAV5-EF1α-DIO-eYFP virus (2.4 × 10$^{13}$ vg mL$^{-1}$) was purchased from the Hope Center Viral Vectors Core at Washington University School of Medicine. The injection was performed with a borosilicate glass capillary pulled to a fine point (50 µm)[74], attached to a Hamilton syringe mounted in a QSI automatic injector (Stoelting). The injector was then attached to a stereotaxic apparatus (Stoelting). The y arm of the stereotaxic apparatus was adjusted to a 45° angle. Animals were anesthetized using ketamine (100 mg kg$^{-1}$, i.p.) and xylazine (15 mg kg$^{-1}$, i.p.) and placed in the stereotaxic apparatus. Following an incision

along the dorsal midline, the L4 and L5 DRG were exposed by removal of the lateral processes of the vertebrae. The epineurium lying over the DRG was opened, and the glass needle inserted into the ganglion, to a depth of 300 µm from the surface of the exposed ganglion. After a 3-min delay to allow sealing of the tissue around the glass capillary tip, 500 nl virus solution was injected at a rate of 30 nl min$^{-1}$. After a further delay of 2 min, the needle was removed. The L4 ganglion was injected first followed by L5. The muscles overlying the spinal cord were loosely sutured together with a 5–0 suture and the wound closed. Animals were then placed in clean cage on paper towel with heated pad for recovery.

**Real-time RT-PCR**. DRG and spinal cords were dissected out on ice and stored in RNA stabilizer (RNAlater, QIAGEN). Total RNA was isolated and genomic DNA was removed in accordance with manufacturer's instructions (RNeasy plus mini kit; QIAGEN). Single-stranded cDNA was synthesized by using High Capacity cDNA Reverse Transcription Kit (Life Technologies). Gene expression of Grp was determined by real-time PCR (StepOnePlus; Applied Biosystems)[46]. Specific primers were designed with the NCBI Primer-BLAST. The primers used are:

Grp: 5′-TGGGCTGTGGGACACTTAAT-3′ (exon1); 5′-GCTTCTAGGAGGT CCAGCAAA-3′; Actb: 5′-TGTTACCAACTGGGACGACA-3′; 5′-GGGGTGTTG AAGGTCTCAAA-3′.

Real-time PCR was carried out with FastStart Universal SYBR Green Master (Roche Applied Science). All samples were assayed in duplicates. Thermal cycling was initiated with denaturation at 95 °C for 10 min. After this initial step, 40 cycles of PCR (heating at 95 °C for 10 s and 60 °C for 30 s) were performed. Data were analyzed using Comparative CT Method (StepOne Software v2.2.2.) and the expression of Grp was normalized to the expression of Actb.

**Retrograde labeling of DRG/TG neurons from skin**. To retrogradely label DRG or TG neurons innervating the hairy nape or cheek skin, the fluorescent tracer 1,1′-Dioctadecyl-3,3,3′,3′-tetramethylindocarbocyanine perchlorate (DiI) (Sigma, 42364) was injected intradermally using a 30-G hypodermic needle. A total of 4–20 µL injections of 5% DiI was made within the nape region on the rostral back skin or 2–20 µL injections into the cheek skin in Grp$^{ChR2}$ mice. After 10 days to allow for labeling, mice were anesthetized, perfused and DRG and TG were dissected and processed for IHC.

**DRG culture and Ca$^{2+}$ imaging**. DRGs from Grp$^{tdTom}$ mice were dissected in neurobasal media (Invitrogen) and incubated with papain (30 µL in 2 mL media; Worthington) for 20 min at 37 °C. After washing with PBS (pH 7.4), cells were incubated with collagenase (3 mg/2 mL media; Worthington) for 20 min at 37 °C. After washing with PBS, cells were dissociated with a flame-polished glass pipette in neurobasal media. Dissociated cells were collected through a cell strainer (BD Biosciences) to remove tissue debris. Dissociated DRG cells were resuspended with DRG culture media (2% fetal bovine serum, 2% horse serum, 2% B-27 supplement, and 1X glutamine in neurobasal media; Invitrogen) and plated onto poly-ornithine–coated 12-mm-diameter round glass coverslips in a 24-well plate. After 16–24 h, cell culture media were replaced with Ca$^{2+}$ imaging buffer (140 mM NaCl, 4 mM KCl, 2 mM CaCl2, 1 mM MgCl2, 5 mM glucose, 10 mM Hepes, adjusted to pH 7.4 with NaOH). Fura-2 acetoxymethyl ester (fura-2 AM) (Invitrogen) was diluted to a 2mM stock in DMSO/20% pluronic acid. The coverslips were mounted on a Warner Instruments recording chamber (RC 26 G) perfused with Ca$^{2+}$ imaging buffer at a rate of ~2 mL/min. An inverted microscope (Nikon Eclipse Ti 10X objective) with Roper CoolSNAP HQ2 digital camera was used for fura-2 Ca$^{2+}$ imaging after 340/380-nm laser excitations (sampling interval, 2 s; exposure time adjusted to ~40 ms for 340 nm and to ~30 ms for 380 nm). CQ (1 mM, Sigma), histamine (50 mM, Sigma), capsaicin (10 or 100 nM, Sigma), or KCl (30 mM, Sigma) was applied to Grp$^{tdTom}$ DRG cultures to examine Ca$^{2+}$ responses. Responsive neurons were identified as ROIs and $F_{340}/F_{380}$ ratios were measured using NIS-Elements (version 3.1, Nikon)[41]. Prism 7 (version 7.0c, GraphPad) was used to analyze Ca$^{2+}$ imaging data.

**Optical skin stimulation behavior**. Grp$^{ChR2}$ mice and wild-type littermates (Grp$^{WT}$) were used for optical skin stimulation experiments. The nape skin was shaved 3 days prior to stimulation in all mice tested. One day prior to the experiments, each mouse was placed in a plastic arena (10 × 11 × 15 cm) for 30 min to acclimate. Mice were videotaped using SONY HDR-CX190 digital video camcorders from a side angle. Mice were recorded for 30 min prior to stimulation for spontaneous scratches. For blue light skin stimulation, a fiber optic cable that was attached to a fiber-coupled 473 nm blue laser (BL473T8-150FC, Shanghai Laser and Optics Co.) with an ADR-800A adjustable power supply. Laser power output from the fiber optic cable was measured using a photometer (Thor Labs) and set to 15 mW from the fiber tip. An Arduino UNO Rev 3 circuit board (Arduino) was programmed and attached to the laser via a BNC input to control the frequency and timing of the stimulation (1, 5, 10, or 20 Hz with 10 ms on-pulse and 3 s On – 3 s Off cycle for 5 min). During stimulation, the mouse was traced manually by a fiber optic cable with ferrule tip that was placed 1–2 cm above the nape skin. Following stimulation, the mice were recorded for 30 min. for spontaneous scratches. Morphine (5 mg kg$^{-1}$, i.p.) was injected 10 min prior to pre-stimulation spontaneous scratching observations. Bombesin-saporin (200 ng/5 µL, i.t.,

Advanced Targeting Systems, Cat No. IT-40) was injected 2 weeks prior to optical stimulation. Videos were played back on a computer for assessments by observers blinded to the animal groups and genotypes. A scratch was defined as a lifting of the hind limb towards the nape or head to scratch and then a replacing of the limb back to the floor, regardless of how many scratching strokes take place between lifting and lowering of the hind limb.

**Western blot**. DRG and lumbar spinal cord of 9-week-old WT and *Grp* cKO mice were dissected on ice and quickly frozen in −80 °C. Soluble cell plasma proteins were extracted as described[75]. Samples were removed into a microtube containing ice-cold sample buffer (20 mM Tris-HCl [pH 7.4], 1 mM dithiothreitol, 10 mM NaF, 2 mM Na3VO4, 1 mM EDTA, 1 mM EGTA, 5 mM microcystin-LR, and 0.5 mM phenylmethylsulfonyl fluoride), and homogenized by sonication. Homogenates were centrifuged at $12,000 \times g$ for 30 min at 4 °C. The supernatant was used for analysis. Protein concentration was determined using BCA assay (Thermo Scientific). For each sample, 10 μg of total protein were separated on SDS NuPAGE Bis-Tris 4–12% gels (Life Technology) in MES running buffer (Life Technology) and transferred to polyvinylidene fluoride membrane (Life Technology). The blots were blocked in blocking buffer (5% nonfat dry milk in PBS and 0.1% Tween 20) for 1 h at room temperature and incubated with rabbit anti GRP (ImmunoStar, 1:5,000), for 16 h at 4 °C. This was followed by 1 h incubation in goat horseradish peroxidase-linked secondary antibodies (Santa Cruz) at 1:2500. Immunoblots were developed with the enhanced chemiluminescence reagents (Amersham).

**Diphtheria toxin receptor-mediated ablation of *Grp* spinal neurons**. To ablate DTR-expressing *Grp* spinal neurons for behavioral and histochemical studies, *Grp*[WT]; *Lbx1*[Flpo]; *Tau*[ds-DTR] control or *Grp*[Cre-KI]; *Lbx1*[Flpo]; *Tau*[ds-DTR] littermates were intraperitoneally injected with diphtheria toxin (DTX, 40 μg kg⁻¹; Sigma Cat. No. D0564) dissolved in saline at day 1 and then again at day 4. Behavioral, ISH and IHC experiments were performed 2 weeks after DTX injection.

**Single-cell qRT-PCR**. Single-cell qRT-PCR was carried out using Ambion® Single Cell-to-CT™ Kit (Life technologies) in accordance with manufactures instructions. Briefly, single tdTomato neuron in laminae I-II of spinal cord slices from *Grp*[tdTom] mice was identified by red fluorescence under microscope. Negative pressure was applied to the pipette to isolate cytosol of the cell, which was extruded into 10 μl cell lysis/Dnase I solution for RNA extraction and genomic DNA digestion. After reverse transcription (25 °C,10 min/42 °C, 60 min/85 °C, 5 min) target cDNA was pre-amplified for 14 cycles (95 °C, 15 s/60 °C, 4 min) in the presence of 0.2x pooled TaqMan assays (ThermoFisher Scientific). Diluted pre-amplification products (1:20 in 1x TE buffer) was used for final qPCR reaction (4 μl, 40 cycles of 95 °C 5 s/60 °C 30 sec; StepOnePlus, Applied Biosystems) to examine target gene expression. TaqMan assays used are: *Actb*, Mm01205647_g1; *Grp* Mm00612997_m1*, Grpr*, Mm01157247_m1; *Nmbr*, Mm00435147_m1; *Npr1*, Mm01220076_g1; *Vglut2*, Mm00499876_m1; *Pdyn*, Mm00457573_m1; *Sst*, Mm00436671_m1; *Npy*, Mm01410146_m1; *Tacr1*, Mm00436892_m1. Data were analyzed using StepOne Software (v2.2.2.) with automatic baseline and threshold was set to 0.2.

**Acute itch behavior**. Behavioral experiments were performed during the day (0800–1500 h)[16]. For injections, mice had their nape shaved a day before the experiments. One day prior to the experiments, each mouse was placed in a plastic arena (10 × 11 × 15 cm) for 30 min to acclimate. On the test day, mice were given at least 10 min to get accustomed to recording conditions prior to injections and recordings. CQ (200 μg, Sigma), SLIGRL-NH2 (100 μg, Genscript), BAM8-22 (100 μg, Genscript), histamine (200 μg, Sigma), or β-alanine (100 μg, Sigma) dissolved in a volume of 50 μL saline was injected intradermally using a syringe attached to a SS30M3009 – 3/10 cc, 30 G × 3/8″ needle (Terumo). Immediately after injections, mice were put into rectangular, transparent observation boxes and videotaped from a side angle. The videos were played back on a computer and quantified by an observer who was blinded to the treatment or genotype. A scratch was defined as a lifting of the hind limb towards the nape or head to scratch and then a replacing of the limb back to the floor, regardless of how many scratching strokes take place between lifting and lowering of the hind limb. Only scratches to the injection site were counted for 30 min.

**Acute pain behavior**. Capsaicin cheek behavior: 1 day prior to injection, the cheek skin was shaved. On the test day, mice were given at least 30 min to get accustomed to recording conditions prior to injections and recordings. Capsaicin (20 μg) dissolved in a volume of 20 μL saline with 2% ethanol and 2% Tween was injected intradermally to the cheek using a syringe with 30 G × 3/8″ needle. Immediately after injections, mice were put into rectangular, transparent observation boxes and videotaped from a side angle. The videos were played back on a computer and quantified by an observer who was blinded to the treatment or genotype. A wipe was defined as a singular motion of one forelimb beginning at the caudal extent of the cheek and proceeding in a rostral direction. Only wipes to the injection site were counted for 30 min.

Thermal sensitivity: thermal sensitivity was determined using hotplate (50, 52, or 56 °C), and Hargreaves assay. For the hotplate test, the latency for the mouse to lick its hindpaw or jump was recorded. For the Hargreaves test, thermal sensitivity was measured using a Hargreaves-type apparatus (IITC Inc.). The latency for the mouse to withdraw from the heat source was recorded.

Mechanical sensitivity: mechanical thresholds were assessed using a set of calibrated von Frey filaments (Stoelting). Each filament was applied 5 consecutive times and the smallest filament that evoked reflexive flinches of the paw on 3 of the 5 trials was taken as paw withdrawal threshold.

Calf injections: for calf injections, mice had their calf shaved a day before the experiments. One day prior to the experiments, each mouse was placed in a plastic arena (10 × 11 × 15 cm) for 30 min to acclimate. On the test day, mice were given at least 10 min to get accustomed to recording conditions prior to injections and recordings. CQ (100 μg), SLIGRL-NH2 (100 μg), or BAM8-22 (100 μg) dissolved in a volume of 10 μL saline was injected intradermally using a syringe attached to a SS30M3009 – 3/10 cc, 30 G × 3/8″ needle (Terumo). Immediately after injections, mice were put back to the plastic arena and videotaped from a side angle. For morphine analgesia experiments, saline (10 μL) or morphine (0.3 nmol in 10 μL saline) was administrated via intrathecal injection. Mice then received calf injection of CQ (100 μg) 30 min later. The videos were played back on a computer and quantified by an observer who was blinded to the treatment or genotype. The time the animal spent on biting/licking toward injection site were counted for 30 min.

**Statistics**. Statistical methods are indicated when used. Values are reported as the mean ± standard error of the mean (SEM). Statistical analyses were performed using Prism 7 (v7.0d, GraphPad, San Diego, CA). For comparison of 1, 5, 10, and 20 Hz stimulation-induced scratching in *Grp*[wt] and *Grp*[ChR2] mice, a one-way ANOVA with Tukey post hoc analysis was performed. For comparison between *Grp*[wt], *Grp*[ChR2], *Grp*[ChR2]-morphine, and *Grp*[ChR2]-BB-Sap behavior analysis during light stimulation, a one-way ANOVA with Tukey post hoc analysis was performed. For comparison between *Grp*[wt], *Grp*[ChR2], *Grp*[ChR2]-morphine, and *Grp*[ChR2]-BB-Sap behavior analysis prior to and following light stimulation, a two-way repeated measures ANOVA with Tukey post hoc analysis was performed. For comparison between *Grp*[F/F] and *Grp*[F/F]; Na$_v$1.8[Cre] mice acute itch behavior, unpaired *t* test was performed. For comparison between *Grp* control and *Grp* spinal ablated dorsal horn neuron numbers of various IHC and ISH markers, unpaired *t* test was performed. For comparison between *Grp* control and *Grp* spinal ablated mice pre- and post-DTX injection, two-way repeated measures ANOVA with Tukey post hoc analysis was performed. Normality and equal variance tests were performed for all statistical analyses. $P < 0.05$ was considered statistically significant.

**Reporting summary**. Further information on research design is available in the Nature Research Reporting Summary linked to this article.

## Data availability
The authors declare that all data supporting the findings of this study are available within the paper and its supplementary information files. The source data underlying Figs. 1h, i, 4d, 5g, 6f–k, h, and 7j–p and Supplementary Figs. 1e, 6a–f, 7c–h, and 8c are provided as a Source Data file.

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

## Acknowledgements

We thank P. Yeager and G. Thompson for technical support and lab members for comments. We also thank M. Goulding and Q. Ma for generously providing *Lbx1*Flpo and *Tau*ds-DTR lines, and X. Dong for kindly helping with generation of floxed GRP mice at Johns Hopkins University School of Medicine Department of Neuroscience Murine Mutagenesis Core. The project has been supported by the NIH grants 1R01AR056318-06, R01NS094344, R01 DA037261-01A1, and R56 AR064294-01A1 (Z.F.C) and the National Science and Technology Major Project of China (2016ZX08011-005) (A.T.).

## Author contributions

Z.F.C. conceived and directed the project. D.M.B., X.Y.L., and Z.F.C. designed the experiments; D.M.B., X.T.L., F.G., S.W., J.L., Q.Y.Y., X.Y.L., X.Z., J.Y., and B.L. performed experiments and analyzed data; A.T. supported the project and discussion. D.M.B., X.Y.L., and Z.F.C. wrote the manuscript.

## Competing interests

The authors declare no competing interests.
