## [Peer Review File · Nature Communications]

Reviewers' comments:

Reviewer #1 (Remarks to the Author):

In this study, Barry et al generated and evaluated a Grp-Cre line, to genetically mark and optogenetically manipulate, Grp-expressing neurons in the dorsal root ganglia, spinal cord, and other regions. While not explicitly mentioned in the study, this line will fate map Grp neurons, and hence may also target neurons that no longer express Grp in adults. Their co-expression analyses suggest that if this is the case, the number of fate mapped neurons is likely to be small.

Overall, this study utilizes a straightforward approach with this new mouse line, and a conditional Grp KO like, to demonstrate that Grp neurons in the DRG, but not spinal cord, contribute to non-histaminergic itch. This study helps to resolve a simmering question in the field as to whether Grp neurons in the DRG and/or spinal cord contribute to itch behaviors.

Reviewer #2 (Remarks to the Author):

Previous studies have shown that neuropeptide GRP plays an important role in itch sensation. However, there are contradicting results on whether GRP is expressed in primary sensory neurons in dorsal root ganglion (DRG). In the current study, Dr. Chen's lab has generated GRP-CreGFP knockin mice and used Cre-GFP dependent marker to demonstrate the presence of GRP in DRG neurons. They further showed that activation of these neurons by optogenetics causes scratch. In addition, DRG specific GRP knockout mice exhibited significantly less scratching than controls. Although the study is potentially interesting, there are some concerns on the quality of the data and inconsistency of the results:

Spinal cord GRP-tdTomato label a few neurons in lamina II (Fig 2G) where GRP in situ shows many positive neurons through dorsal horn (Fig S7). This inconsistency raises the concerns on the quality of GRP in situ. Also, GRP RNAscope showed many more positive cells in the DRG than anti-GRP antibody staining (Fig S1b and Fig 1E). The authors should perform double staining of GRP and Cre to demonstrate the overlap.

GRP-tdTomato labeled more neurons in adult DRG than GRP antibody staining (Fig 3A, 3C versus Fig 1E, 6C). These data suggest there is a broader GRP expression in the DRG during development. The authors should carefully examine GRP expression and GRP-tdTomato in the DRG at earlier time points before adult.

Although GRP-tdTomato labels many DRG neuron cell bodies, there is not much tdTomato positive central terminal fibers in dorsal horn (Fig 2G,2I). Only the cell bodies of spinal cord GRP+ neurons and their dendrites and axons can be seen in these figures. This is very puzzling.

In addition, the central terminal projection of GRP neurons in the spinal cord determined by GRP antibody staining (largely lamina I and II outer shown in Fig 1H and 1I) is inconsistent with the subtypes of GRP-tdtomato labeled neurons in the DRG. As the DRG staining indicated in Fig 3A and 3C, GRP-tdtomato labeled neurons are IB4 positive and CGRP weak (not strong) positive neurons whose central axons should terminate at lamina II inner (not lamina I and IIouter).

It is hard to conceive that a localized injection of CQ is able to induce homogeneous reduction of GRP staining in the dorsal horn (from medial to lateral, Fig 1H ipsi).

The authors should use cheek injection assay to determine whether activation of GRP+ neurons lead to itch (scratching) or pain (wiping) by GRP-DREADD mice.

Reviewer #3 (Remarks to the Author):

While it is well established that gastrin releasing peptide (GRP) is an itch-specific transmitter, the origin and site of action of GRP in the itch pathway are controversial. This study tackles this controversy by using a series of genetic, pharmacological, behavioral and calcium imaging approaches to investigate whether GRP that is expressed in neurons of dorsal root ganglia (DRG) or of the dorsal horn (DH) of the spinal cord mediate scratching in mice. The investigators generated novel mouse lines to identify and characterize GRP expression and to determine the role of GRP in itch sensation. This study further clarifies previous data by the same group and strengthens the argument for the role of GRP expression and release from DRG sensory neurons in itch transmission. The experiments are well described and planned. It makes an important contribution to our understanding of the neural pathways and mediators of chronic itch. Several issues require attention.

1. There are numerous spelling and grammatical errors. The entire manuscript should be carefully proof read.
2. Further studies are required to characterize and validate the reporter mice. In figure 2 g-h, tdTomato expression does not overlap with GRP labeling in the DH, whereas tdTomato expression almost perfectly overlaps with GRP expression in the DRG. Since tdTomato is used to identify GRP fibers, how do the authors explain this difference? Especially since later tdTomato expression is used to characterize GRP-positive neurons.
3. Other studies have questioned the specificity of the rabbit anti-GRP antibody. In DRG, the labeling of GRP neurons matches the labeling of AAV-DIO-eYFP. However, no labeling is shown in the DH, even though later the study shows tdTomato-positive neurons in the DH. If the antibody is specific, then why does it only seem to label processes in the DH and not neuronal somas that are tdTomato positive. Also, in figure 6 the western blots show that the antibody has non-specific labeling. What is the evidence that this antibody is selective for GRP over another related neuropeptide? Along the same line, it is important to show that the GRP antibody does not label neurons in the GRP KO mice. RNAscope was used to show loss of RNA but no IHC labeling was shown to demonstrate antibody specificity.
4. In figure 3 the authors characterize GPR neurons but switch between TG and WT mice. Would like to see a comparison of all sensory neuron markers with both TG and WT.
5. There is some inconsistency in the figures in that the n values are not always indicated – e.g., Fig. 1h. I suggest indicating the exact n values for every experiment rather than a range.
6. In Fig 4c, at what time after agonist addition is the snapshot of the Ca level shown? Without this information, it is difficult to interpret and compare these data.
7. In Fig 4d, is each line a trace from an individual neuron?

We would like to thank the reviewers for their very positive and constructive comments, which have guided the careful revision of the manuscript. To address the concerns, we have performed additional experiments and made the changes in the figures and text accordingly. Notably, we have generated *Grp*-DREADD mice and performed chemogenetic activation of GRP fibers using the cheek model and found mice scratch the cheek with minimal wiping behavior. Together, the present study represents the most comprehensive demonstration of the presence of itch-dedicated primary afferents in mice. The following is the point-by-point response. Note that the changes in MS are marked in **Red** for reading convenience, except Introduction and Discussion, which have been extensively revised.

Reviewer #1

In this study, Barry et al generated and evaluated a *Grp*-Cre line, to genetically mark and optogenetically manipulate, *Grp*-expressing neurons in the dorsal root ganglia, spinal cord, and other regions. While not explicitly mentioned in the study, this line will fate map *Grp* neurons, and hence may also target neurons that no longer express *Grp* in adults. Their co-expression analyses suggest that if this is the case, the number of fate mapped neurons is likely to be small.

Overall, this study utilizes a straightforward approach with this new mouse line, and a conditional *Grp* KO like, to demonstrate that *Grp* neurons in the DRG, but not spinal cord, contribute to non-histaminergic itch. This study helps to resolve a simmering question in the field as to whether *Grp* neurons in the DRG and/or spinal cord contribute to itch behaviors.

A: We would like to thank the reviewer for comments.

Reviewer #2

Previous studies have shown that neuropeptide GRP plays an important role in itch sensation. However, there are contradicting results on whether GRP is expressed in primary sensory neurons in dorsal root ganglion (DRG). In the current study, Dr. Chen's lab has generated GRP-CreGFP knockin mice and used Cre-GFP dependent marker to demonstrate the presence of GRP in DRG neurons. They further showed that activation of these neurons by optogenetics causes scratch. In addition, DRG specific GRP knockout mice exhibited significantly less scratching than controls. Although the study is potentially interesting, there are some concerns on the quality of the data and inconsistency of the results:

1. Spinal cord GRP-tdTomato label a few neurons in lamina II (Fig 2G) where GRP in situ shows many positive neurons through dorsal horn (Fig S7). This inconsistency raises the concerns on the quality of GRP in situ.

A: Thanks for the comments. Seemingly inconsistent images are due to different magnification and the RNAscope staining. For example, Fig. 2G is a high-power image which only shows a few GRP-tdTomato neurons (for comparison, please see a low-power image (Fig. S2C)).

2. Also, GRP RNAscope showed many more positive cells in the DRG than anti-GRP antibody staining (Fig S1B and Fig 1E). The authors should perform double staining of GRP and Cre to demonstrate the overlap.

A: The apparent discrepancy of the two images is due to the different magnifications and techniques used. The high-power image in Fig. 1E showed fewer positive cells, whereas a low-power image of anti-

GRP antibody staining (Fig. 6C) showed comparable number of positive cells as the GRP RNAscope image (Fig. S1B). The anti-GRP antibody (Fig. 2D-F) and AAV5-DIO-eYFP virus (Fig. 1E-G) were used to label GRP-Cre neurons in the DRGs. Unfortunately, various anti-CRE antibodies could not produce specific immunostaining signal, and we are unable to find a good Cre antibody that can be used for double IHC staining with GRP antibody. Nevertheless, our virus transfection result provides direct evidence to show the activity of Cre in DRGs (Fig. 1F).

3. GRP-tdTomato labeled more neurons in adult DRG than GRP antibody staining (Fig 3A, 3C versus Fig 1E, 6C). These data suggest there is a broader GRP expression in the DRG during development. The authors should carefully examine GRP expression and GRP-tdTomato in the DRG at earlier time points before adult.

A: GRP antibody staining largely overlapped with Grp-tdTomato in the DRG (Fig. 2D-F). The number of positive cells in Fig. 6C (GRP antibody staining) is comparable to that in Fig. S8A (GRP-tdTomato) and in Fig. S1B (GRP RNAscope). tdTomato staining in DRGs sometimes varies depending on the sections and segmental level of the spinal cord. We replaced Fig. 3A and 3C with more representative images.

4. Although GRP-tdTomato labels many DRG neuron cell bodies, there is not much tdTomato positive central terminal fibers in dorsal horn (Fig 2G,2I). Only the cell bodies of spinal cord GRP+ neurons and their dendrites and axons can be seen in these figures. This is very puzzling.

Fig. 1. a, b, Representative sections from postnatal day 30 (P30) *Tac1^{cre}-tdTomato* mice ($n = 3$), showing tdTomato (red) with *Tac1* mRNA (green) or NK1R (green) in superficial dorsal spinal laminae. Arrows indicate co-localization and arrowheads indicate singular expression. Inset in **b** shows a NK1R⁺tdTomato⁺ cell. (Huang et al., 2019)

A: We do not have a good explanation for lack of fiber staining in the dorsal horn. However, it is not a rare observation for tdTomato mouse strains. For example, *Tac1-Cre/tdTomato*¹ (Fig. 1 in this rebuttal letter) and *Sst-Cre/tdTomato*² labeled very few primary afferents in the dorsal horn.

5. In addition, the central terminal projection of GRP neurons in the spinal cord determined by GRP antibody staining (largely lamina I and II outer shown in Fig 1H and 1I) is inconsistent with the subtypes of GRP-tdtomato labeled neurons in the DRG. As the DRG staining indicated in Fig 3A and 3C, GRP-tdtomato labeled neurons are IB4 positive and CGRP weak (not strong) positive neurons whose central axons should terminate at lamina II inner (not lamina I and IIouter).

A: Also see the response to #4. We replaced Fig. 3A and 3C with more representative images. The majority of Grp^{tdTom} -labeled neurons (71%, 54 of 76) in the DRG are CGRP positive. About 26% of Grp^{tdTom} -labeled neurons (33 of 127) are IB4 positive. Several explanations can account for some inconsistencies between GRP IHC and tdTomato staining in DRGs. It is possible that GRP protein, once it is synthesized, is rapidly transported to the terminals, resulting in fewer GRP positive cell bodies that can be visualized by GRP IHC. It is always easier to visualize GRP central terminals in the spinal cord using GRP antibody. Further, GRP IHC can also be influenced by several factors, such as the area examined, the time of perfusion (GRP expression in DRG is under major circadian regulation with highest expression in the afternoon, lowest in the early morning, *unpublished data*) and how the animal tissue is processed etc. Even in SCN where GRP antibody is one of the best antibodies (along with AVP and VIP) used widely in the circadian field for decades³, it could be difficult to detect GRP cell bodies in the core^{4, 5}, unless SCN is treated with colchicine⁶. Some technical caveats about GRP antibody have been previously discussed in detail⁷. Also we cannot exclude the possibility with certainty that there may be some developmental lineage of tdTomato in DRGs. In this study, these issues do not affect our conclusion regarding the role of GRP in DRGs and GRP fibers as itch fibers, which are based on functional studies.

6. It is hard to conceive that a localized injection of CQ is able to induce homogeneous reduction of GRP staining in the dorsal horn (from medial to lateral, Fig 1H ipsi).

A: This experiment was inspired by our previous finding that in SCN, GRP IHC staining is markedly reduced after mice watched scratching video for 30 min⁵. To maximize the release of GRP from central terminals more broadly, a total of three sites were selected for CQ injection in order to cover most of the area of the right nape so that the mouse scratched one side only and the contralateral side can be used as the control (see Fig. 2a in this

letter). IHC staining confirmed that Fos⁺ cells were found throughout the superficial dorsal horn from medial to lateral region of the injected side (Fig. 2b in this rebuttal letter). The fact that animals scratch all the time even without pruritogenic stimuli necessitates a mechanism operated in central terminal vesicles to ensure the

Fig. 2. a, Diagram showing the injection sites for CQ on the mouse nape skin. **b,** Representative image of c-Fos staining on the cervical spinal section of a mouse injected with CQ.

spontaneous release of GRP onto the spinal cord. As an internal control, we included SP IHC image (Fig. 1i).

7. The authors should use cheek injection assay to determine whether activation of GRP+ neurons lead to itch (scratching) or pain (wiping) by GRP-DREADD mice.

A: Done. We have bred GRP-Cre-eGFP line with R26-LSL-Gq-DREADD line to generate mice with hM3Dq expression in Grp neurons (Grp^{Gq}). After injection of clozapine-N-oxide into the cheek skin, Grp^{Gq} mice showed robust scratching behaviors towards the injection site. In contrast, Grp^{WT} mice barely showed any scratching responses due to lacking Gq DREADD (Supplementary Figure 6e). Importantly, Grp^{WT} and Grp^{Gq} mice showed minimal wiping behaviors (Supplementary Figure 6f).

Reviewer #3

While it is well established that gastrin releasing peptide (GRP) is an itch-specific transmitter, the origin and site of action of GRP in the itch pathway are controversial. This study tackles this controversy by using a series of genetic, pharmacological, behavioral and calcium imaging approaches to investigate whether GRP that is expressed in neurons of dorsal root ganglia (DRG) or of the dorsal horn (DH) of the spinal cord mediate scratching in mice. The investigators generated novel mouse lines to identify and characterize GRP expression and to determine the role of GRP in itch sensation. This study further clarifies previous data by the same group and strengthens the argument for the role of GRP expression and release from DRG sensory neurons in itch transmission. The experiments are well described and planned. It makes an important contribution to our understanding of the neural pathways and mediators of chronic itch.

Several issues require attention.

1. There are numerous spelling and grammatical errors. The entire manuscript should be carefully proof read.

A: Thanks for helpful comments, and the revised ms has been carefully proof-read.

2. Further studies are required to characterize and validate the reporter mice. In figure 2 g-h, tdTomato expression does not overlap with GRP labeling in the DH, whereas tdTomato expression almost perfectly overlaps with GRP expression in the DRG. Since tdTomato is used to identify GRP fibers, how do the authors explain this difference? Especially since later tdTomato expression is used to characterize GRP-positive neurons.

A: Please see the response to reviewer 2.

3. Other studies have questioned the specificity of the rabbit anti-GRP antibody. In DRG, the labeling of GRP neurons matches the labeling of AAV-DIO-eYFP. However, no labeling is shown in the DH, even though later the study shows tdTomato-positive neurons in the DH. If the antibody is specific, then why does it only seem to label processes in the DH and not neuronal somas that are tdTomato positive. Also, in figure 6 the western blots show that the antibody has non-specific labeling. What is the evidence that this antibody is selective for GRP over another related neuropeptide? Along the same line, it is important to show that the GRP antibody does not label neurons in the GRP KO mice. RNAscope was used to show loss of RNA but no IHC labeling was shown to demonstrate antibody specificity.

A: Please also see the response to the reviewer 2. Lack of GRP IHC labeling in the DRG and the spinal cord of GRP CKO mice indicates that GRP antibody is specific for IHC. Our data suggest that Grp neurons in the dorsal horn may not express GRP peptide. However, we always found residual GRP fiber staining in the dorsal horn in both KO mice and after dorsal root rhizotomy, which most likely reflects descending projection of GRP in the brainstem region. Some argue that GRP may be like SP peptide that is expressed in the dorsal horn, but rapidly release, making it difficult to see GRP-positive cell bodies. While this possibility certainly exists, we did not find any GRP staining in dorsal horn neuronal culture *in vitro*⁷. Nevertheless, the evidence presented in this study shows that spinal *Grp* neurons are not required for itch nor pain transmission, regardless of whether GRP peptide is expressed endogenously in the spinal cord.

The non-specific labeling of western blots is likely due to the sample treatments that denatured the proteins and exposed some epitopes that were not readily accessible for IHC labeling. It is worth mentioning that the specificity of GRP antibody was previously verified using global GPR KO mice (Zhao et al. 2013 and Fig. 6C-F)⁸. For your reference, the GRP IHC staining in sensory neurons of rats was replicated beautifully in rats by another lab, indicating that GRP is conserved in rodents⁹.

4. In figure 3 the authors characterize GPR neurons but switch between TG and WT mice. Would like to see a comparison of all sensory neuron markers with both TG and WT.

A: We repeated the experiments and performed double immunostaining of GRP/CGRP, GRP/IB4 and GRP/TRPV1 with WT (Fig. 3a-f). For Mrgpra3 and Hrh1, we cannot perform immunostaining because specific antibodies are not available. We tried RNAscope using Mrgpra3 and Hrh1 probes on DRG sections of GRP-tdTomato mice. However, the tdTomato signals failed to survive the RNAscope procedures. We have to performed double in situ hybridization of Grp/Mrgpra3 and Grp/Hrh1 on DRG sections from WT mice (Fig. 3g and i).

5. There is some inconsistency in the figures in that the n values are not always indicated – e.g., Fig. 1h. I suggest indicating the exact n values for every experiment rather than a range.

A: Thanks for the suggestion. The exact n values are added for every experiment.

6. In Fig 4c, at what time after agonist addition is the snapshot of the Ca level shown? Without this information, it is difficult to interpret and compare these data.

A: The snapshots were taken 20 s after agonist addition. The information is added to Fig. 4C.

7. In Fig 4d, is each line a trace from an individual neuron?

A: Yes.

1 Huang, T. *et al.* Identifying the pathways required for coping behaviours associated with sustained pain. *Nature* **565**, 86-90, doi:10.1038/s41586-018-0793-8 (2019).

2 Chamessian, A. *et al.* Transcriptional Profiling of Somatostatin Interneurons in the Spinal Dorsal Horn. *Sci Rep-Uk* **8**, doi:ARTN 6809

10.1038/s41598-018-25110-7 (2018).

3 Yan, L. *et al.* Exploring spatiotemporal organization of SCN circuits. *Cold Spring Harb Symp Quant Biol* **72**, 527-541, doi:10.1101/sqb.2007.72.037 (2007).

4 Welsh, D. K., Takahashi, J. S. & Kay, S. A. Suprachiasmatic nucleus: cell autonomy and network properties. *Annu Rev Physiol* **72**, 551-577, doi:10.1146/annurev-physiol-021909-135919 (2010).

5 Yu, Y. Q., Barry, D. M., Hao, Y., Liu, X. T. & Chen, Z. F. Molecular and neural basis of contagious itch behavior in mice. *Science* **355**, doi:10.1126/science.aak9748 (2017).

6 Karatsoreos, I. N., Yan, L., LeSauter, J. & Silver, R. Phenotype matters: identification of light-responsive cells in the mouse suprachiasmatic nucleus. *J Neurosci* **24**, 68-75, doi:10.1523/JNEUROSCI.1666-03.2004 (2004).

- 7 Barry, D. M. *et al.* Critical evaluation of the expression of gastrin-releasing peptide in dorsal root ganglia and spinal cord. *Mol Pain* **12**, doi:10.1177/1744806916643724 (2016).
- 8 Zhao, Z. Q. *et al.* Chronic itch development in sensory neurons requires BRAF signaling pathways. *J Clin Invest* **123**, 4769-4780, doi:10.1172/JCI70528 (2013).
- 9 Takanami, K. *et al.* Distribution of gastrin-releasing peptide in the rat trigeminal and spinal somatosensory systems. *The Journal of comparative neurology* **522**, 1858-1873 (2014).

REVIEWERS' COMMENTS:

Reviewer #1 (Remarks to the Author):

I had no comments as I had reviewed this paper at a previous journal and the reviewers addressed my initial comments in the first submission to Nat. comm.

Reviewer #2 (Remarks to the Author):

The authors have addressed most of the reviewer's concerns. They should include brief response to the reviewer's concern #4 (lacking of tdtomato projections in the spinal cord) and #5 (GRP project pattern in the spinal cord) in the revised manuscript.

Reviewer #3 (Remarks to the Author):

Thank you for thoroughly addressing my concerns. I have no further recommendations for revision.

REVIEWERS' COMMENTS:

Reviewer #1 (Remarks to the Author):

I had no comments as I had reviewed this paper at a previous journal and the reviewers addressed my initial comments in the first submission to Nat. comm.

Reviewer #2 (Remarks to the Author):

The authors have addressed most of the reviewer's concerns. They should include brief response to the reviewer's concern #4 (lacking tdtomato projections in the spinal cord) and #5 (GRP project pattern in the spinal cord) in the revised manuscript.

A: Thanks for the suggestion. We add two sentences in the revised manuscript (see page 5).

Reviewer #3 (Remarks to the Author):

Thank you for thoroughly addressing my concerns. I have no further recommendations for revision.